# An introduction to large deviations with applications in physics

**Ivan N. Burenev[1], Daniël W. H. Cloete[2], Vansh Kharbanda[3,4] and Hugo Touchette[2]***

**1** Laboratoire de physique théorique et modèles statistiques (LPTMS),
Université Paris-Saclay, France
**2** Department of Mathematical Sciences, Stellenbosch University, South Africa
**3** Faculty of Physics and Center for NanoScience, LMU München, Germany
**4** Department of Veterinary Sciences, Institute for Infectious Diseases and Zoonoses,
LMU München, Germany

* htouchette@sun.ac.za

*Part of the 2024-07: Theory of Large Deviations and Applications collection*
*Session 123 of the Les Houches School, July 2024*
*published in the Les Houches Summer School Lecture Notes series*

## Abstract

These notes are based on the lectures that one of us (HT) gave at the Summer School on the "Theory of Large Deviations and Applications," held in July 2024 at Les Houches in France. They present the basic definitions and mathematical results that form the theory of large deviations, as well as many simple motivating examples of applications in statistical physics, which serve as a basis for the many other lectures given at the school that covered more specific applications in biophysics, random matrix theory, nonequilibrium systems, geophysics, and the simulation of rare events, among other topics. These notes extend the lectures, which can be accessed online, by presenting exercises and pointer references for further reading.

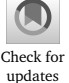

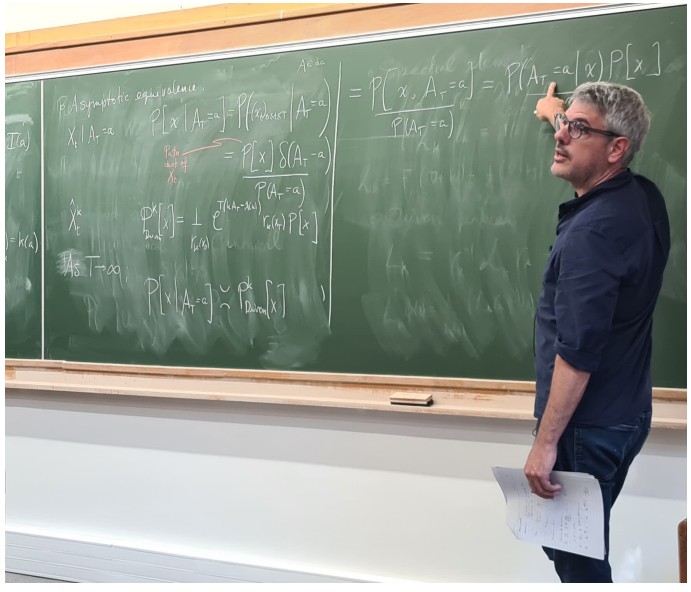

# 1 Introduction

The first of the two summer schools organised in 2024 at the Les Houches School of Physics was on the topic of large deviation theory and its applications in mathematics and physics. The following notes are based on the initial set of lectures given by Hugo at that school, introducing the basic concepts and results of that theory, used in many of the lectures that followed on its applications for studying rare events and fluctuation phenomena arising in nonequilibrium physics, geophysics, biophysics, quantum physics, as well as more theoretical areas, such as random matrix theory. The notes are not a transcript of the lectures, as such, but go through all the material that was presented in each of the six lectures, sometimes with some minor changes of notations, and try to capture the message that was conveyed in each of them. We arranged the notes by topics rather than by lectures and also added pointers for exercises and further reading.

The video recordings of the lectures can be found on the channel [1] hosted by the Université de Grenoble Alpes for the École de Physique des Houches. The lectures themselves follow many references and material used in previous courses and schools, including Hugo's 2009 review paper on the large deviation approach to statistical mechanics [2], notes for a course on large deviations given at Stellenbosch University [3], and sets of lecture notes written for two summer schools [4,5]. The latter sources are used in particular for exercises.

# 2 Basics of large deviation theory

This first section lays the ground for the applications covered in the later sections and, more widely, for the other lectures given at the school by defining the core problem of large deviation theory – that is, to determine the probability of rare events, fluctuations or large deviations arising in systems composed of many components – and by introducing the basic tools and results used to address this problem. We start with a number of motivating applications and examples to give a sense of how rare events arise in different systems and how the probability of such events follows a general scaling or approximate form, formalised in the theory by the large deviation approximation or principle. After defining this principle, we explain how the probabilities of large deviations are calculated in practice using the Gärtner–Ellis theorem, and then spend some time discussing the properties of Legendre transforms, which underlie this theorem and much of large deviation theory.

## 2.1 Applications of large deviations

The following are examples of situations or problems dealing with large deviations. Some will be studied again in later sections, while others are covered in other lectures, so we do not explain them in full detail at this point.

- **Sample means of IID random variables:** An example that will be used repeatedly in the notes to explain some points of the theory about large deviations is a sample mean having the form

$$S_n = \frac{1}{n}\sum_{i=1}^{n} X_i \,, \tag{1}$$

where $X_1, \ldots, X_n$ are *random variables* (RVs). In the simplest case, we take these to be independent and to have the same distribution, and so the RVs are said to be IID for *independent* and *identically distributed*. Even with this simplification, it is not easy to find the distribution $P(S_n = s)$ of the sample mean. However, in many cases, it is possible to

find the following scaling approximation:

$$P(S_n = s) \approx e^{-nI(s)}, \tag{2}$$

which is valid as $n \to \infty$ or, in practice, when $n$ is large. We will discuss later the meaning of the approximation sign $\approx$ and how large $n$ must be for this approximation to be effective. For now, we note that, in this form, the distribution decays exponentially with the number of RVs with a speed or rate determined by a function $I(s)$, called the *rate function*.

- **Additive observables of Markov processes:** The RVs in the sample mean above do not need to be IID for the approximation in (2) to arise. In a more general way, we can consider them to be correlated by assuming that they form a Markov chain in discrete time or a Markov process $(X_t)_{t\geq0}$ evolving in continuous time. In the latter case, the discrete sample mean $S_n$ is replaced by the time-average

$$S_T = \frac{1}{T} \int_0^T f(X_t) \, dt, \tag{3}$$

where $f$ is some function of the process $X_t$ and $T$ is the integration time up to which that process is observed. Under some conditions on $X_t$ and $f$, we find in the limit $T \to \infty$,

$$P(S_T = s) \approx e^{-TI(s)}, \tag{4}$$

with a rate function $I(s)$ that depends on the process and function considered.

This type of approximation will be considered again in Sec. 5 and is important in physics when studying the fluctuations of nonequilibrium systems modelled as Markov processes. In this context, $X_t$ represents the system that we study, while $S_T$ is some quantity or *observable* that we measure over time.

- **Particle systems at equilibrium:** Large deviation theory was first applied in physics to study the fluctuations of many-particle systems at equilibrium with a fixed energy (microcanonical ensemble) or with a heat bath with a fixed temperature (canonical ensemble) [2, 6]. In this context, the system is a collection of $N$ particles whose state is the list $\omega = ((x_1, p_1), \ldots, (x_N, p_N))$ of coordinates $x_i$ and momenta $p_i$. In statistical physics, this state is treated as a list or vector of RVs, distributed according to the Gibbs distribution:

$$P_\beta(\omega) = \frac{e^{-\beta H_N(\omega)}}{Z(\beta)}. \tag{5}$$

This holds for a system in contact with a heat reservoir and involves the inverse temperature $\beta$ of the reservoir, as well as the Hamiltonian or energy function $H_N(\omega)$ of the $N$ particles.

Since the energy depends on the state, it is an RV on its own, so we can try to determine its distribution. This is a difficult problem, in general, but a simplification arises if we consider the thermodynamic limit $N \to \infty$. Then the distribution of the energy per particle $h_N = H_N/N$ takes the approximate form

$$P(h_N = u) \approx e^{-NI_\beta(u)}. \tag{6}$$

Therefore, as before, we have that the distribution of an observable decays exponentially with the number of components in the system – here the number of particles – with a rate function $I_\beta(u)$ that depends on the value or fluctuation $H_N = uN$ considered and the inverse temperature of the bath (as a parameter).

This type of approximation will not be considered in the lectures or indeed at the school, though it is quite important theoretically and historically because of its many connections with thermodynamics. We refer to [2] for more details.

- **Nonequilibrium systems:** The framework just described can be generalised to study the fluctuations of observable quantities in systems driven out of equilibrium by external forces or boundary reservoirs injecting energy or particles. The study of these systems is an active area in statistical physics, covered at the school in the lectures of Bernard Derrida, Pablo Hurtado, and Vivien Lecomte.

  A typical application considered in this context is that of a system, say a metal rod, in contact at its ends with a hot and a cold bath or environment. Because of the temperature difference, a heat current will flow through the rod (Fourier's law), which appears to us to be constant at the macroscopic level. However, if we were to measure that current at a more microscopic level, then fluctuations would be observed, which turn out to be distributed according to

$$P(J_{T,V} = j) \approx e^{-TVI(j)}. \tag{7}$$

  Here, $J_{T,V}$ represents the current averaged over a time $T$ and over a volume $V$ and the approximation is valid in the combined limit where $VT \to \infty$. As before, the rate of decay is determined by a rate function $I(j)$, whose form depends on the system considered. This is covered in Derrida's and Hurtado's lectures with particle-based models of heat and particle transport processes, which can be studied at the particle level or at a more macroscopic level using the macroscopic or hydrodynamic fluctuation theory.

- **Rare transitions in noise-perturbed systems:** Here is a different system for which an exponentially decaying probability also appears. We imagine a particle with state $X_t \in \mathbb{R}^d$ evolving in a potential $U : \mathbb{R}^d \to \mathbb{R}$ according to a gradient descent dynamics and add a small Gaussian noise to the evolution, so as to obtain the following stochastic differential equation:

$$dX_t = -\nabla U(X_t)\, dt + \sqrt{\varepsilon}\, dW_t\,, \tag{8}$$

  where $W_t$ is a Brownian motion in $\mathbb{R}^d$ modelling the noise and $\varepsilon$ is the variance of the noise. For this model, we are interested in finding the probability that $X_t$ goes from some attractor (i.e., a minimum) of $U(x)$, labelled by $A$, to another attractor $B$, separated from $A$ by a potential barrier. Without noise, this transition is impossible: starting in the basin of attraction of $A$, $X_t$ would "fall" into $A$ and similarly for $B$. However, with noise, it is possible for $X_t$ to go from $A$ to $B$ by climbing the potential barrier between them. As $\varepsilon \to 0$, it is known that the probability of this event scales according to

$$P(A \to B) \approx e^{-I(A \to B)/\varepsilon}\,, \tag{9}$$

  where $I(A \to B)$ is now a number that depends on the potential considered.

  This type of transition is covered in the lectures of Eric Vanden-Eijnden. It is important in chemistry for describing thermally-activated transitions between metastable states (e.g., folding states of complex proteins), in nonequilibrium physics for describing the appearance of fluctuations in the macroscopic limit (see the lectures of Pablo Hurtado and of Baruch Meerson), and in geophysics for describing sudden transitions or extreme events in climate dynamics and fluid dynamics (see the lectures of Freddy Bouchet).

- **Random matrices:** Exponentially-decaying probabilities similar to those seen so far arise in many other contexts, whether in physics, science in general, or in mathematics. One important application, coming from random matrix theory, considers a square matrix $A$ of size $N$ with symmetric entries $A_{ij} = A_{ji}$ distributed independently according to

a Gaussian distribution with mean 0 and variance 1 (or 2 for the diagonal elements). This defines the so-called Gaussian orthogonal ensemble of matrices, characterised by real eigenvalues $\lambda_i$, $i = 1, \ldots, N$, whose joint distribution can be found explicitly.

For this probabilistic model, we can look at the fraction

$$F_N = \frac{1}{N} \sum_{i=1}^{N} \theta(\lambda_i - a) \tag{10}$$

of eigenvalues lying above some threshold $a$, $\theta(x)$ being the Heaviside function. This is another RV, playing the role of observable for a random matrix, whose distribution is approximated as

$$P(F_N = f) \approx e^{-N^2 I(f)}, \tag{11}$$

in the limit where $N \to \infty$. Results of this type are discussed in the lectures of Mylène Maïda and Pierpaolo Vivo, who cover more generally the topic of random matrices from the mathematical and spin glass viewpoints, respectively.

## 2.2 Simple examples of IID sample means

The exponential approximation of the probability distributions found in all the previous applications is the focus of large deviation theory, which we will get to define more precisely in the next section as the large deviation approximation or *large deviation principle*. In a simple way, large deviation theory is a theory that predicts when this approximation arises and provides techniques for obtaining the exponent or rate function describing the exponential decay of a given probability distribution with some parameter (e.g., the number of random variables in a sample, the number of particles in system, etc.). Before we get to the definition of the large deviation principle, we consider next two simple examples of sample means to better understand the meaning of the exponential decay and the approximation as such.

**Example 1.** We consider again a sample mean of IID RVs

$$S_n = \frac{1}{n} \sum_{i=1}^{n} X_i, \tag{12}$$

presented as our first application, and suppose now that $X_i$ are Gaussian RVs with a mean $\mu$ and a variance $\sigma^2$. This is an easy case to consider because it is known that a sum of Gaussian RVs is Gaussian-distributed, so we can write directly without much calculation

$$P(S_n = s) = \sqrt{\frac{n}{2\pi\sigma^2}} \exp\left[-n\frac{(s-\mu)^2}{2\sigma^2}\right]. \tag{13}$$

The plot of this probability distribution, a probability density really, is shown in Fig. 1 for a number of increasing value of $n$. As expected, $P(S_n = s)$ concentrates on the mean $\mu$ as $n$ increases, which is a consequence of the law of large numbers: in the limit where $n \to \infty$, $S_n$ concentrates in probability to its mean, corresponding here to $E[S_n] = \mu$. The source of that concentration can be seen in the result above as coming from the exponential term, which is dominant compared to the $\sqrt{n}$ term, so we can write

$$P(S_n = s) \approx e^{-nI(s)}, \tag{14}$$

to leading order in $n$ with the rate function

$$I(s) = \frac{(s-\mu)^2}{2\sigma^2}. \tag{15}$$

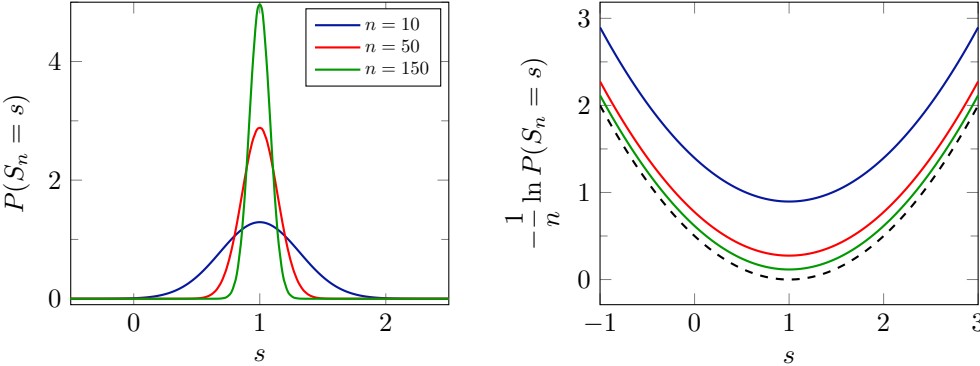

Figure 1: Sample mean of Gaussian random variables with $\mu = 1$ and $\sigma = 1$. Left: Probability density $P(S_n = s)$. Right: Convergence of the LDP limit to the rate function $I(s)$ (dashed curve).

This function is also plotted in Fig. 1 (dashed curve in the right plot). It is strictly positive for $s \neq \mu$, implying that $P(S_n = s)$ decays exponentially with $n$ for those values, and is such that $I(\mu) = 0$, so the value $s = \mu$ is the only value for which $P(S_n = s)$ does not decay exponentially. By conservation of probability, it must then concentrate there, in accordance with the law of large numbers.

**Example 2.** Suppose now that the $X_i$ are Bernoulli RVs taking values $\{0, 1\}$ with probability $P(X_i = 1) = p$ and $P(X_i = 0) = 1 - p$ with $p$ a parameter in $[0, 1]$. The probability distribution of the sample mean $S_n$ can also be found explicitly for this example: the sum itself is distributed according to the binomial distribution, so with the additional $n$ factor we find

$$P(S_n = s) = \frac{n!}{(ns!)(n(1-s))!} p^{ns}(1-p)^{(1-s)n} . \tag{16}$$

To extract the dominant, exponential term of this distribution, we use Stirling's approximation for the factorial, $n! \approx n^n e^{-n}$, obtaining after a few calculations

$$P(S_n = s) \approx e^{-nI(s)} , \tag{17}$$

in the limit $n \to \infty$, where

$$I(s) = s \ln \frac{s}{p} + (1-s) \ln \frac{1-s}{1-p} . \tag{18}$$

This rate function is different from the one found for the Gaussian sample mean, though it is also strictly positive for $s \neq p$ and equal to 0 for $s = p$, which corresponds to the mean of $S_n$. This recovers again the law of large numbers. The exponential approximation makes this result more precise by showing that the concentration is exponential with $n$.

## 2.3 Large deviation principle

Distributions that decay exponentially with some parameters appear in many applications, as seen before, so it makes sense to name this form and to make it more precise with a definition. The idea, as explained in the previous section, is that a distribution $P(S_n = s)$ might have a complicated form as a function of $s$ and $n$, but in the cases that interest us the observation or expectation is that its dominant part is simple: it is a decaying exponential in $n$ characterised by a decay or rate function $I(s)$ that does not depend on $n$. Consequently, we can write

$$P(S_n = s) = e^{-nI(s) + o(n)} , \tag{19}$$

using the small-o notation, $o(n)$, to show that all the corrections to the dominant exponential part are sub-exponential with $n$ and thus strictly smaller than $n$ in the exponential as $n \to \infty$. Taking the logarithm then gives

$$\ln P(S_n = s) = -nI(s) + o(n), \tag{20}$$

so the dominant term is now linear in $n$. To make the small-o term disappear, we then divide by $n$ and take the limit $n \to \infty$ to obtain

$$\lim_{n \to \infty} -\frac{1}{n} \ln P(S_n = s) = I(s) - \lim_{n \to \infty} \frac{o(n)}{n} = I(s). \tag{21}$$

This gives us a way to extract the rate function, as we did in the right plot of Fig. 1. But, more importantly, it gives us a precise meaning of the approximation sign $\approx$ that we used before, leading to our first definition.

**Definition 1 (Intuitive LDP).** *Let $S_n$ be a continuous RV with a probability density $P(S_n = s)$.[1] We say that $S_n$ or $P(S_n = s)$ satisfies the large deviation principle[2] (LDP) if the limit*

$$\lim_{n \to \infty} -\frac{1}{n} \ln P(S_n = s) = I(s), \tag{22}$$

*exists and yields a function $I(s)$ that is not identically $0$ or $\infty$ everywhere.*

In the remaining, we use the sign $\asymp$ instead of $\approx$ to emphasise the asymptotic nature of the exponential approximation. Thus, we write

$$P(S_n = s) \asymp e^{-nI(s)}, \tag{23}$$

to mean that the log limit in (22) exists and gives the rate function $I(s)$. This is a common symbol used in mathematics and information theory, among other topics, related to the notion of asymptotic or logarithmic equivalence. In a nutshell, two sequences of numbers $(a_n)_{n \geq 1}$ and $(b_n)_{n \geq 1}$ are said to be asymptotically equivalent if

$$\lim_{n \to \infty} \frac{1}{n} \ln \frac{a_n}{b_n} = 0. \tag{24}$$

This means, similarly to the LDP, that $a_n$ is equal to $b_n$ up to sub-exponential terms in $n$, and so we write $a_n \asymp b_n$.

The definition of the LDP above is precise enough for most practical applications, but has two shortcomings mathematically: it assumes that the density of $S_n$ exists and that the limit itself exists. In mathematics, there is a preference for characterising an RV not by its distribution or density, but by the probability that it takes values in some arbitrary interval or set. This leads to a slightly more general definition of the LDP stated next.

**Definition 2 (More mathematical LDP).** *The RV $S_n$ or, more precisely, the sequence or family of RVs $(S_n)_{n > 0}$ satisfies the LDP if there exists a function $I(s)$ such that, for any set B,*

$$\lim_{n \to \infty} -\frac{1}{n} \ln P(S_n \in B) = \min_{s \in B} I(s). \tag{25}$$

*As before, we also require $I(s)$ to be different from $0$ or $\infty$.*

---

[1] For simplicity, we denote probability densities with $P$ like probabilities. The context will always be clear as to whether we deal with a probability or a probability density.

[2] Here, the term principle does not refer to a fundamental law of physics or observation, but rather to a definition of an approximation, in the same sense as the Laplace principle used for approximating an integral.

How is this definition related to the first one? First, note that, if $S_n$ has a probability density, we can write

$$P(S_n \in [s, s + \Delta s]) = P(S_n = s)\Delta s, \tag{26}$$

by choosing $B$ to be the infinitesimal interval $[s, s + \Delta s]$. Taking the LDP limit in (25) then yields

$$\lim_{n \to \infty} -\frac{1}{n} P(S_n \in [s, s + \Delta s]) = \lim_{n \to \infty} -\frac{1}{n} \ln P(S_n = s) - \frac{1}{n} \ln \Delta s = I(s), \tag{27}$$

for any small but finite $\Delta s$. In this way, the first definition of the LDP that we stated can be seen as a local version of the second definition involving an arbitrary set $B$.

Another way to relate the two definitions of the LDP is to write, for any $B$,

$$P(S_n \in B) = \int_B P(S_n = s)ds \asymp \int_B e^{-nI(s)}ds, \tag{28}$$

having used the LDP approximation for the last expression. At this point, we notice that an integral of exponential terms can be approximated by its largest integrand and that the error incurred in this approximation is also sub-exponential with $n$, just like in the LDP, so we obtain

$$P(S_n \in B) \asymp \exp\left[-n \min_{s \in B} I(s)\right]. \tag{29}$$

This explains why a minimisation appears in (25) in the second definition of the LDP.

The approximation of an exponential integral by the integrand having the maximum exponent is called the *Laplace approximation*, the Laplace principle or the saddle-point approximation (in the context of integrals in the complex plane). It plays a fundamental role in large deviation theory, so it is worth stating explicitly:

$$\int_B e^{-nI(x)}dx \asymp \max_{x \in B} e^{-nI(x)} = e^{-n \min_{x \in B} I(x)}. \tag{30}$$

There is also a Laplace approximation or principle for sums of exponentials:

$$\sum_i e^{-a_i} \asymp \max_i e^{-a_i} = e^{-\min_i a_i}. \tag{31}$$

The asymptotic sign $\asymp$ means again that the two sides of the approximation are logarithmically equivalent, that is, equal up to sub-exponential correction terms in $n$.

The second definition of the LDP is better for a mathematician than the first, but still assumes that the limit exists for any $B$. In large deviation theory, this assumption is dropped by defining the LDP with an upper and a lower limit, called a limsup and liminf, respectively. We state this definition next for the mathematically-inclined readers.

**Definition 3** (**Complete LDP**). *The sequence of RVs $(S_n)_{n>0}$ satisfies the LDP if there exists a function $I(s)$ such that, for any set $B$,*

$$\liminf_{n \to \infty} -\frac{1}{n} \ln P(S_n \in \bar{B}) \geq \inf_{s \in \bar{B}} I(s), \tag{32}$$

*where $\bar{B}$ denotes the closure of $B$, which is a closed set, and*

$$\limsup_{n \to \infty} -\frac{1}{n} \ln P(S_n \in B°) \leq \inf_{s \in B°} I(s), \tag{33}$$

*where $B°$ denotes the interior of $B$, which is an open set.*

We will not use this version of the LDP in these notes, but it is important to show it nonetheless, since it is the definition found in most mathematical articles and books on large deviation theory. For an explanation of all the terms involved in this definition, see Sec. 3.2 and Appendix B of [2]. We also provide some references in the "Further reading" section for more details.

## 2.4 Varadhan's lemma

There are many results in large deviation theory that can be used to determine whether a given RV satisfies the LDP and, if so, to find its rate function. The one that we will focus on in these notes (and lectures) is the Gärtner–Ellis theorem, stated in the next section. To motivate this result, we describe in this section a related result known as Varadhan's lemma, which is a consequence of the LDP. As before, we consider a real RV $A_n$ and an expectation of that RV having the form

$$E[e^{nf(A_n)}] = \int_{\mathbb{R}} e^{nf(a)} P(A_n = a) \, da \,, \tag{34}$$

where $f$ is some function of $A_n$ that makes the expectation finite. The reason for considering this form of expectation is that, if $P(A_n = a)$ satisfies the LDP with a rate function $I(a)$, then

$$E[e^{nf(a)}] \asymp \int_{\mathbb{R}} e^{n[f(a) - I(a)]} \, da \,. \tag{35}$$

We recognize again a Laplace integral, which we approximate to exponential order as

$$E[e^{n(f(a))}] \asymp e^{n \max_a [f(a) - I(a)]} \,. \tag{36}$$

To extract the exponent, we then follow the LDP by defining the limit

$$\lambda[f] = \lim_{n \to \infty} \frac{1}{n} \ln E\left[e^{nf(A_n)}\right] \,, \tag{37}$$

to obtain in the end

$$\lambda[f] = \max_{a \in \mathbb{R}} \{f(a) - I(a)\} \,. \tag{38}$$

We put $f$ in square brackets to emphasise that the limit depends on the function $f$ and so is a functional of $f$.

Varadhan's lemma is a generalisation of this result that is applicable to a large class of RVs, not just real RVs with a probability density, including random vectors and even random functions. We state it next and explain after how we can use it to obtain the rate function.

**Theorem 1** (**Varadhan 1966**). *Let $(A_n)_{n>0}$ be a sequence of RVs defined on some (nice) space $\mathcal{A}$ and satisfying the LDP with a (nice) rate function $I(a)$. For any real and bounded function $f : \mathcal{A} \to \mathbb{R}$ of $A_n$, we have*

$$\lambda[f] = \lim_{n \to \infty} \frac{1}{n} \ln E\left[e^{nf(A_n)}\right] = \sup_{a \in \mathcal{A}} \{f(a) - I(a)\} \,. \tag{39}$$

*The* sup *in this expression is a generalised maximum that can return $\infty$.*

The word "nice" above refers to technical conditions that are beyond the scope of these notes; see the references in the Further reading section for more details. For the purpose of these notes, what is important to note is that Varadhan's lemma generalises the Laplace principle to general and abstract RVs satisfying the LDP. At a more practical level, the lemma also shows a way to obtain the rate function $I(a)$. To see this, take $A_n$ to be a real RV and choose $f$ to be the linear function $f(a) = ka$ with $k \in \mathbb{R}$. Although this is not a bounded function of $A_n$, we are allowed to use it in Varadhan's lemma provided that the limit in (37) defining $\lambda[f]$ is finite. In this case, we obtain a function of $k$, expressed as $\lambda(k)$, which must be such that

$$\lambda(k) = \sup_{a \in \mathbb{R}} \{ka - I(a)\} \,, \tag{40}$$

according to Varadhan's lemma. This is an interesting result for two reasons:

- The maximisation in (40) has the form of a Legendre transform, known more precisely as a *Legendre–Fenchel transform*.

- Legendre transforms can be inverted, under some conditions on $\lambda(k)$ or $I(a)$, to obtain

$$I(a) = \sup_{k \in \mathbb{R}} \{ka - \lambda(k)\}. \tag{41}$$

The Gärtner–Ellis theorem is related to the second point: it provides a sufficient condition on $\lambda(k)$ for $I(a)$ to be the Legendre transform of this function, without knowing $I(a)$ or even that $A_n$ satisfies the LDP.

## 2.5 Gärtner–Ellis theorem

The version of the Gärtner–Ellis (GE) theorem that we state is actually a simplified version close to the theorem first proved by Gärtner [7] and later generalised by Ellis [8]. Following the previous section, we define the function

$$\lambda(k) = \lim_{n \to \infty} \frac{1}{n} \ln E\left[e^{nkA_n}\right], \tag{42}$$

where $k \in \mathbb{R}$ and $A_n$ is at this point a real RV not known a priori to satisfy the LDP (we will consider cases later where $A_n$ is a random vector). This function is called the *scaled cumulant generating function* (SCGF) of $A_n$, since the exponential expectation of $A_n$ is essentially the generating function of $A_n$ and the log of a generating function is called a cumulant. In terms of the SCGF, we have the following.

**Theorem 2 (Gärtner 1977, Ellis 1984).** *If $\lambda(k)$ is finite and differentiable in a neighbourhood of $k = 0$, then*

- *$A_n$ satisfies the LDP:*

$$P(A_n = a) \asymp e^{-nI(a)}. \tag{43}$$

- *The rate function $I(a)$ is the Legendre–Fenchel (LF) transform of the SCGF:*

$$I(a) = \sup_{k \in \mathbb{R}} \{ka - \lambda(k)\}. \tag{44}$$

There are two important points to note about this result:

- The nature of $A_n$ is unspecified at this point; in particular, $A_n$ need not be a sample mean or involve IID RVs. From the physics point of view, this means that the GE theorem can be applied in principle to any observable of systems that involve correlated or uncorrelated particles, so long as the SCGF can be calculated. Indeed, the difficulty of obtaining a rate function for any system comes from the difficulty of obtaining its SCGF.

- Rate functions obtained from the GE theorem are necessarily convex (in fact, strictly convex) because they result from the LF transform, which produces by definition only convex functions. This does not mean that rate functions are always convex. We will see later that the SCGF is always convex by definition, but there is nothing in the definition of a rate function that says that it has to be convex. If a rate function is nonconvex (we will see an example in the next section), then it cannot be obtained from the GE theorem.

We will explore these points in detail in these notes. Already we can check that the GE theorem recovers the rate function of the Gaussian IID sample mean studied before. In this case, we have

$$E[e^{nkS_n}] = \prod_{i=1}^{n} E[e^{kX_i}] = E[e^{kX_1}]^n, \tag{45}$$

as a result of the independence of the RVs and the fact that they are identically distributed. The SCGF, consequently, has the simple form

$$\lambda(k) = \ln E[e^{kX_1}], \tag{46}$$

which is the cumulant of $X_1$ or any of the other RVs in $S_n$, since they are identically distributed. From here, we then use the Gaussian distribution to find that the cumulant is

$$\lambda(k) = \ln \int_{\mathbb{R}} e^{kx} p(x) dx = \mu k + \frac{\sigma^2}{2} k^2. \tag{47}$$

This is finite and differentiable for all $k \in \mathbb{R}$, so we can take the LF transform to find

$$I(s) = k_s s - \lambda(k_s) = \frac{(s - \mu)^2}{2\sigma^2}, \tag{48}$$

where $k_s$ is the unique root of $\lambda'(k) = s$. This recovers the result found in Example 1.

The advantage of this calculation, as can be appreciated, is that we obtain the rate function with a few simple steps that involve no probabilities – only elements of analysis for calculating an integral and the LF transform. This is the advantage of the GE theorem and large deviation theory more generally: rather than attempting to calculate the full distribution of observables, which in most cases is very difficult (if at all possible), we settle for the easier (yet still challenging) task of calculating the dominant part of that distribution with techniques that often involve very little probability theory. The price paid in doing so is not so great because the dominant part is exponential, and so contributes to the full distribution in an overwhelming way.

## 2.6 Legendre–Fenchel transform

The fact that the rate function and SCGF are related by a LF transform is a fundamental result in large deviation theory, having many consequences and ramifications in physics, in particular for understanding why Legendre transforms appear in thermodynamics (see Sec. 5 of [2]). We review here some properties of this transform, considering for simplicity real functions of one variable. In this context, the LF transform takes a function $f : \mathbb{R} \to \mathbb{R}$ and transforms it to another function given by

$$f^*(k) = \sup_x \{kx - f(x)\}. \tag{49}$$

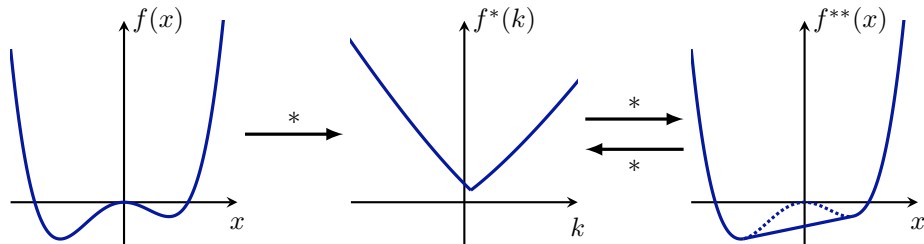

Figure 2: Properties of the LF transform $f^*$ of a function $f$ and the double LF transform $f^{**}$.

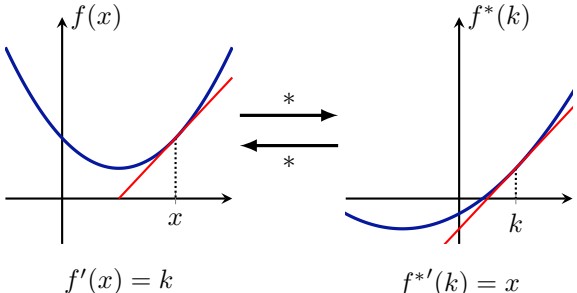

$$f'(x) = k \qquad\qquad f^{*\prime}(k) = x$$

Figure 3: Duality between the slopes and values of a function $f$ and its LF transform $f^*$.

By re-applying the transform to $f^*(k)$, we obtain the double LF transform:

$$f^{**}(x) = \sup_k \{kx - f^*(k)\}. \tag{50}$$

These two functions have several important properties that we state without proofs:[3]

- The function $f^*(k)$ is always convex.

- $f^{**}(x)$ represents in general the convex envelope of $f(x)$ (see Fig. 2).

- If $f(x)$ is convex, then $f^{**}(x) = f(x)$, so applying the LF transform twice gives $f$ back. We say in this case that the LF transform is involutive.

- If $f(x)$ is not convex, then $f(x) \geq f^{**}(x)$ (see Fig. 2).

- When $f$ is convex, the values and slopes of $f$ and $f^*$ are related by duality in the following sense. Consider a point $x$ such that $f'(x) = k$. Then $f^{*\prime}(k) = x$. This is illustrated in Fig. 3. In short, we say that the slopes of $f$ map to the points of $f^*$, while the slopes of $f^*$ maps to the points of $f$.

- If $f(x)$ is strictly convex and differentiable, then the maximisation involved in the LF transform can be solved exactly by finding the unique root of

$$\frac{\partial}{\partial x}[kx - f(x)] = 0. \tag{51}$$

This results in

$$f^*(k) = kx_k - f(x_k), \tag{52}$$

where $x_k$ is the unique solution of $f'(x) = k$. This is the expression of the standard Legendre transform, so for strictly convex and smooth functions, the LF transform reduces to the Legendre transform. We will see in the next section that this result, which we used already in (48), can be combined with the duality to obtain the rate function in a parametric way without having to solve for a root.

---

[3]Some of the results are valid except possibly at points on the boundary of the domain of $f$. This is a technical issue in convex analysis that we do not go into.

## 2.7 Properties of $\lambda(k)$ and $I(a)$

We close this section by listing some general properties of the SGCF and rate function, leaving their proof as exercises.

- $\lambda(0) = 0$. This follows from the definition of this function.

- $\lambda(k)$ is by definition a convex function of $k$.

- If $I(a)$ is convex, then it is the LF transform of $\lambda(k)$. This is not quite the GE theorem because the condition presupposes some knowledge about $I(a)$. In the GE theorem, we only need to know $\lambda(k)$, specifically whether $\lambda(k)$ is differentiable, which is a sufficient (but not necessary) condition for obtaining $I(a)$ as the LF transform of $\lambda(k)$.

- The derivative of $\lambda(k)$ at $k = 0$ is the asymptotic expectation:

$$\lambda'(0) = \lim_{n\to\infty} E[A_n] = a^*. \tag{53}$$

  This value determines in general a zero of $I(a)$ and thus a concentration point of $P(A_n = a)$, which is unique in many cases. For IID sample means, for example, we have $\lambda'(0) = \mu$, the mean itself.

- The second derivative of $\lambda(k)$ at $k = 0$ gives the asymptotic variance:

$$\lambda''(0) = \lim_{n\to\infty} n\mathrm{Var}[A_n]. \tag{54}$$

  This implies that the variance of $A_n$ decreases like $1/n$, which shows in a different way that $P(A_n = a)$ is concentrating as $n \to \infty$. If $I(a)$ has a unique zero at $a^*$, then we have by duality

$$\lambda''(0) = I''(a^*)^{-1}, \tag{55}$$

  so the curvature of $I(a)$ around its minimum determines the variance of $A_n$.

## 2.8 Exercises

- Recap on probability theory: [9] and Week 1 material from the course [3].

- Basic examples of LDPs: Week 2 material from [3] and Exercises 2.7.1-2.7.5 in [4].

- Varadhan's lemma and the GE theorem: Week 3 material from [3] and Exercises 2.7.6-2.7.10 in [4].

- Rate function of IID sample means: See next section.

## 2.9 Further reading

- Applications of large deviations in physics: [2, 12, 13].

- Varadhan's lemma: Sec. 4.3 of [14] and Sec. III.3 of [15].

- Gärtner–Ellis theorem: Sec. 2.3 of [14] and Sec. V.2 of [15].

- Accessible mathematical books on large deviation theory: [14, 15].

- LF transforms and convex analysis: [10, 11].

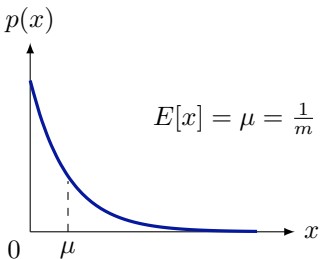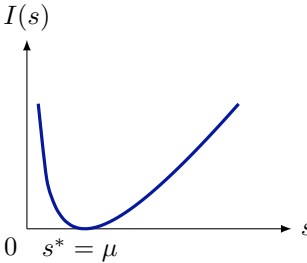

Figure 4: Left: Probability density of an exponentially-distributed RV. Right: Rate function of the corresponding sample mean.

## 3 Simple applications

The aim in this section is to get some practice into calculating rate functions by applying the GE theorem to the simplest observable possible, namely, sample means of IID RVs. In this context, the GE theorem recovers a result by Cramér [16] dating from 1938, considered by mathematicians to be the first large deviation result. The examples that we cover are simple but important to get an idea of the kinds of rate functions that are possible, what can be inferred from the knowledge of these functions, and how the duality between rate functions and SCGFs operates in practice. At the end, we also consider an example of vector sample mean, related to another important result in large deviation theory, known as Sanov's theorem, to explain how this type of RV is covered by the GE theorem.

### 3.1 Cramér's theorem

We have seen already that the application of the GE theorem to a sample mean

$$S_n = \frac{1}{n} \sum_{i=1}^{n} X_i \tag{56}$$

of IID RVs yields a simple form of SCGF, namely,

$$\lambda(k) = \ln E\left[e^{kX}\right], \tag{57}$$

where $X$ stands for any of the RVs in $S_n$ (they are all the same). Thus, provided that the SCGF of $X$ exists and is differentiable, $P(S_n = s)$ satisfies the LDP and so concentrates exponentially with $n$ according to a rate function $I(s)$ given by the LF transform of $\lambda(k)$.

This result is known as Cramér's theorem following the work of Cramér [16], who studied for the first time the distribution of IID sample means beyond the central limit theorem, thereby initiating the subject of large deviations.[4] We study next some applications to better understand the form and properties of $I(s)$.

**Example 3.** As a variation of the Gaussian sample mean studied before, take the $X_i$'s to be distributed according to an exponential distribution having the form

$$p(x) = \begin{cases} me^{-mx}, & x \geq 0, \\ 0, & x < 0, \end{cases} \tag{58}$$

---

[4]Harald Cramér (1893-1985) was a statistician working in actuarial mathematics. His interest in large deviations came from trying to study the distributions of insurance claims.

where $m > 0$ is a parameter related to the expectation or mean of $X$ according to $E[X] = \mu = 1/m$. For this density, shown in Fig. 4, the SCGF is found to be

$$\lambda(k) = \ln \frac{m}{m-k} = -\ln(1 - \mu k), \tag{59}$$

which is differentiable for $k < 1/\mu = m$. Taking the Legendre transform then gives

$$I(s) = \frac{s}{\mu} - 1 - \ln \frac{s}{\mu}, \qquad s > 0. \tag{60}$$

This is an interesting form of rate function:

- As for the rate function of the Gaussian sample mean, $I(s)$ is convex, positive, and has a zero at the mean $s^* = \mu$, which corresponds to the concentration point of $P(S_n = s)$. This confirms again that $S_n \to \mu$ in probability in the limit $n \to \infty$, in agreement with the law of large numbers.

- The concentration is now skewed, since $I(s)$ is asymmetric, as seen in Fig. 4. For $s \gg \mu$, $I(s) \sim s/\mu = sm$, which means that $P(S_n = s) \approx e^{-nms}$ and thus that the large fluctuations – the large deviations – of $S_n$ are exponentially distributed just like the RVs themselves. For $s \to 0$, the behaviour is different: $I(s) \sim \ln \mu/s$, implying with the LDP, $P(S_n = s) \approx 1/s^n$ up to multiplicative and additive constants.

- Near the mean, $I(s)$ is locally quadratic, so the small fluctuations or deviations of $S_n$ around $s^* = \mu$ are Gaussian with a variance given by $\lambda''(0)/n = \mu^2/n$. This follows essentially from the central limit theorem.

The second and third properties explain the term "large deviations": the rate function describes both the small (generally Gaussian) fluctuations of observables as well as their large (generally non-Gaussian) fluctuations. In this sense, the LDP can be seen as an extension of the central limit theorem giving information beyond the Gaussian regime of fluctuations.

**Example 4.** Now we take the RVs in the sample mean $S_n$ to be discrete Bernoulli RVs taking values in $\{0, 1\}$ with probability $P(X_i = 0) = 1 - p$ and $P(X_i = 1) = p$, as we did in Example 2. Instead of obtaining the LDP from the knowledge of $P(S_n = s)$, as we did in that example, we now use Cramér's theorem. The SCGF in this case is

$$\lambda(k) = \ln \left[ p e^k + (1 - p) \right], \tag{61}$$

and is differentiable for all $k \in \mathbb{R}$ (see Fig. 5). Therefore, we can take its LF transform to obtain the rate function, resulting in

$$I(s) = s \ln \frac{s}{p} + (1 - s) \ln \frac{1 - s}{1 - p}, \tag{62}$$

for $s \in [0, 1]$. From this result, we note:

- $I(s)$ is convex, positive, and its zero is located at the mean $E[X] = p$. This makes sense: $S_n$ counts the fraction of 1's in a given sequence, which tends to converge to $p$, the probability used for generating the 1's.

- $I(s)$ is defined only on $[0, 1]$ because $S_n$ represents again the fraction of 1's in a sequence of Bernoulli trials. Outside of this range, we set $I(s) = \infty$, since $P(S_n = s) = 0$ for $s \notin [0, 1]$.

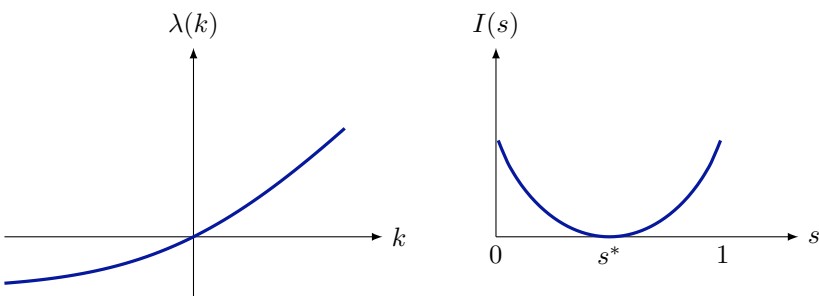

Figure 5: SCGF (left) and rate function (right) for the sample mean of $n$ Bernoulli RVs, plotted for $p = 1/2$.

- $I(1) = -\ln p$ in agreement with the fact that $P(S_n = 1) = p^n$. Similarly, $I(0) = -\ln(1-p)$ because $P(S_n = s) = (1-p)^n$. In both cases, the rate function gives the exact probability without corrections.

- $I(s)$ is a continuous function of $s$ even though $S_n$ is discrete for any finite $n$. This arises because $I(s)$ is the limiting function that describes the set of values (fluctuations) of $S_n$ as $n \to \infty$, which becomes dense in $[0, 1]$ in that limit. Another reason is that $I(s)$ gives the probability of any event $B$ according to our second definition of the LDP. Thus, for any $B$ containing discrete values of $S_n$, we have

$$P(S_n \in B) \asymp e^{-n \min_{s \in B} I(s)}, \tag{63}$$

so $I(s)$ is evaluated correctly at one of those values. The point of having a continuous rate function is that the minimisation above can be evaluated for any $B$ and any $n$.

**Example 5.** We can repeat the calculations of the previous examples using any distribution (discrete or continuous) of our choosing. One example that is interesting from a computational point of view is the uniform distribution on the unit interval

$$p(x) = \begin{cases} 1, & x \in [0, 1], \\ 0, & \text{otherwise.} \end{cases} \tag{64}$$

The cumulant of this distribution is

$$\lambda(k) = \begin{cases} \ln \dfrac{e^k - 1}{k}, & k \neq 0, \\ 0, & k = 0, \end{cases} \tag{65}$$

and is differentiable for all $k \in \mathbb{R}$. Therefore, as before, we can compute its Legendre transform to obtain $I(s)$. However, unlike the previous examples, the calculation of that transform cannot be done exactly or symbolically because the equation $\lambda'(k) = s$ leads to a transcendental equation. In this case, we can compute numerically the rate function in two ways:

- **Direct numerical way:** Solve $\lambda'(k) = s$ numerically to obtain the root $k_s$ (using any root finding routine) and repeat this procedure on a grid of points $s$ to obtain the rate function on that grid as

$$I(s) = k_s s - \lambda(k_s). \tag{66}$$

- **Parametric way:** Use the duality to express the rate function as

$$I(s_k) = k s_k - \lambda(k),  \tag{67}$$

as a function of $k$, where $s_k = \lambda'(k)$. In this way, we choose a grid of values for $k$, calculate the corresponding values $s_k = \lambda'(k)$ of the observable (here, the sample mean), and finally compute the rate function for these values with the formula above. The advantage compared with the previous method is that there is no time spent finding roots – we only need to evaluate $\lambda(k)$ and its derivative on a grid of points.

We leave it as an exercise to try these two methods for the uniform distribution. The rate function in this case is defined on $[0,1]$, naturally, and is symmetric around the mean $s^* = \mu = 1/2$.

**Example 6.** We illustrate in this example a point we made before, namely, that rate functions can be nonconvex even though the SCGF is always convex (by definition). To this end, we consider a variation of the Gaussian sample mean, referred to as the mixed Gaussian sample mean, in which the $X_i$'s are Gaussian RVs with mean 0 and variance 1, and add to that sample mean a new discrete RV $Y \in \{-1, 1\}$, so as to get

$$S_n = Y + \frac{1}{n} \sum_{i=1}^{n} X_i.  \tag{68}$$

For simplicity, we assume that $Y$ is independent of the $X_i$'s and choose $P(Y=-1)=P(Y=1)=\frac{1}{2}$.

To find the distribution of $S_n$, we condition on the value of $Y$ to write

$$
\begin{aligned}
P(S_n = s) &= P(S_n = s | y = -1) P(y = -1) + P(S_n = s | y = 1) P(y = 1) \\
&= \frac{1}{2} P(S_n = s | y = -1) + \frac{1}{2} P(S_n = s | y = 1),
\end{aligned}  \tag{69}
$$

and note that, conditioned on $Y = \pm 1$, $S_n$ is a sample mean of Gaussian RVs translated by $\pm 1$. Thus,

$$P(S_n = s | Y = \pm 1) \asymp e^{-n I_\pm(s)},  \tag{70}$$

where

$$I_\pm(s) = \frac{(s \pm 1)^2}{2},  \tag{71}$$

is the rate function of a sample mean of IID Gaussian RVs translated according to the value of $Y$. Inserting this in (69) then gives

$$P(S_n = s) \asymp \frac{1}{2} e^{-n I_-(s)} + \frac{1}{2} e^{-n I_+(s)} \asymp e^{-n \min\{I_-(s), I_+(s)\}},  \tag{72}$$

the last approximation following from the Laplace principle. Consequently, $P(S_n = s)$ satisfies the LDP with the rate function

$$I(s) = \min\{I_-(s), I_+(s)\}.  \tag{73}$$

This function has two minima, as shown in Fig. 6, and is nonconvex as a result. Therefore, it cannot be obtained from the GE theorem. To understand what would happen if we tried to apply this theorem, let us calculate the SCGF. From the definition of $S_n$, we have

$$\lambda(k) = \lim_{n \to \infty} \frac{1}{n} \ln E[e^{nkY} e^{k \sum_{i=1}^{n} X_i}] = \lim_{n \to \infty} \frac{1}{n} \ln E[e^{nkY}] + \ln E[e^{kX}],  \tag{74}$$

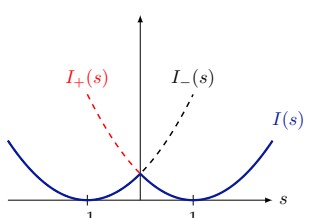
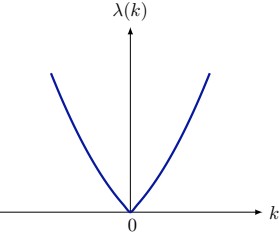
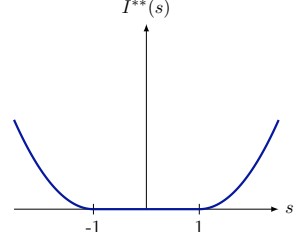

Figure 6: Left: Nonconvex rate function (in blue) of the mixed Gaussian sample mean. Middle: Associated SCGF which is non-differentiable at $k = 0$. Right: LF transform of $\lambda(k)$ giving the convex envelope $I^{**}(s)$ of $I(s)$.

having used the fact that $Y$ is independent of the $X_i$'s and that the $X_i$'s themselves are IID. The second term in this expression is the cumulant of a normal RV; as for the first term, it can be checked that

$$\lim_{n\to\infty} \frac{1}{n} E[e^{nkY}] = \lim_{n\to\infty} \frac{1}{n} \ln \frac{e^{-nk} + e^{nk}}{2} = |k|\,. \tag{75}$$

Therefore, we find

$$\lambda(k) = |k| + \frac{k^2}{2}\,. \tag{76}$$

From this result, we see that we cannot apply the GE theorem because $\lambda(k)$ is non-differentiable at $k = 0$, as shown in Fig. 6. To be sure, we can calculate the LF transform of $\lambda(k)$ above (try as an exercise), but then we get not $I(s)$ but its convex envelope $I^{**}(s)$, as also shown in Fig. 6.

Interestingly, if we take the Legendre transform of either differentiable branches $\lambda(k)$ below and above $k = 0$, we do recover the branches of $I(s)$ that are equal to $I^{**}(x)$. In this sense, the GE theorem can be applied locally to points of $\lambda(k)$ that are differentiable. What the example shows, however, is that the nonconvex region of $I(s)$ for which $I(s) > I^{**}(x)$ cannot be obtained as the LF or Legendre transform of $\lambda(k)$. That region is responsible for the non-differentiability of $\lambda(k)$. This is a general result of convex analysis (see Sec. 4.4 of [2] and [11] for more details).

## 3.2 Sanov's theorem

The problem considered by Sanov [18] is the following. We are given a sequence of $n$ IID RVs $X_1,\ldots,X_n$ taking values in a discrete set, say $\{1, 2,\ldots,q\}$, and consider the sample mean

$$L_{n,j} = \frac{1}{n} \sum_{i=1}^{n} \delta_{X_i,j}\,, \tag{77}$$

which gives the fraction of RVs in the sequence having the value $j$. In the lectures, the case of two values ($q = 2$) labelled by 0 and 1 is considered, so $X_1,\ldots,X_n$ is a binary sequence containing a certain fraction $L_{n,0}$ of 0's and a complementary fraction $L_{n,1}$ of 1's. Putting these two numbers together in a list, we obtain a vector $L_n$ corresponding to the histogram of the values appearing in a given sequence. The problem is to determine the distribution of this vector, which is also called the *empirical vector* in large deviation theory.

Sanov did not have the benefit of the GE theorem to find that the distribution $P(L_n = l)$ has an exponential form with $n$. Instead, he proceeded similarly as we did for the sample mean of Bernoulli RVs (Example 2) by noting that the exact distribution of $L_n$ is the multinomial distribution and by applying the Stirling approximation to this distribution to find

$$P(L_n = l) \asymp e^{-nD(l||p)}\,, \tag{78}$$

where

$$D(l||p) = \sum_{i=1}^{q} l_i \ln \frac{l_i}{p_i} \tag{79}$$

is the rate function involving the components $l_i$ of the vector $l$ and the probability $p_i$ of each value $1, 2, \ldots, q$. This rate function is also called the relative entropy or Kullback–Leibler distance of $l$ relative to $p$. Interestingly, Boltzmann did the same calculation by considering the occupation fractions of a gas of particles having a discrete number of energies and arrived at the same result, 80 years before Sanov (see [17] for more details).

With the hindsight of the GE theorem, we can re-derive Sanov's result in a more direct way without knowing the exact distribution of $L_n$. To this end, we need to take the parameter $k$ entering in the SCGF to be a vector having the same dimension as $L_n$. Since the latter is a sample mean of IID RVs, corresponding in fact to Bernoulli RVs, we find for the SCGF

$$\lambda(k) = \ln E[e^{k \cdot \delta_X}] = \ln \sum_{i=1}^{q} p_i \, e^{\sum_j k_j \delta_{i,j}} = \ln \sum_{i=1}^{q} p_i e^{k_i} \,. \tag{80}$$

This is a differentiable function of $k$ in the vector sense. As a result, we can use the GE theorem to obtain the rate function $I(l)$ of $L_n$ as the vector Legendre transform of $\lambda(k)$:

$$I(l) = \sup_{k} \{k \cdot l - \lambda(k)\}, \tag{81}$$

where $k \cdot l = \sum_{i=1}^{q} k_i l_i$ is the standard scalar product of $k$ and $l$. We leave the computation of this Legendre transform as an exercise; the result is the relative entropy above, so that $I(l) = D(l||p)$.

**Example 7.** For a binary sequence of IID RVs, the relative entropy is

$$D(l||p) = l_0 \ln \frac{l_0}{p_0} + l_1 \ln \frac{l_1}{p_1} \,, \tag{82}$$

where $l_0$ and $l_1$ are the fractions of 0's and 1's, respectively, and $p_0$ and $p_1$ their corresponding probabilities. By normalisation, $l_0 = 1 - l_1$ and $p_1 = 1 - p_0$, so we can also write

$$D(l||p) = (1-s) \ln \frac{1-s}{1-p} + s \ln \frac{s}{p} \,, \tag{83}$$

using $l_1 = s$ and $p_1 = p$. This explains the form of the rate function found in Example 2 for the Bernoulli sample mean.

Note that, since $D(l||p)$ can be obtained from a Legendre transform, it must be a convex function of $l$, which is also positive. Its unique zero is $l = p$, which confirms the observation (backed by the law of large numbers) that the fraction of times the value $i$ appears in a very long sequence of IID RVs is very likely to be close to $p_i$. This is the typical value of $L_{n,i}$, so $p$ is by extension the typical value of $L_n$ as a vector. In the binary case, for example, the typical fraction of 1's in a long sequence of 0's and 1's is $p$, the probability used to generate the 1's. The LDP expresses this observation more quantitatively by showing that any sequence containing a fraction of 1's departing from $p$ is exponentially unlikely to appear with the length of the sequence.

## 3.3 Exercises

- IID sample means and Sanov's theorem: Exercises 3.6.1, 3.6.8, and 3.6.9 of [4].

- Extra: Exercises 3.6.2-3.6.7 of [4].

### 3.4 Further reading

- Cramér's theorem: Original article: [16]; historical account: [17]; more modern presentation: Sec. 2.2 of [14] and Sec. I.3 of [15].

- Nonconvex rate functions: Sec. 4.4 of [2] and [11].

- Sanov's theorem: Original article: [18]; presentation: Week 4 material from the course [3], Sec. 3.1.2 of [14], and Sec. II.1 of [15].

- Contraction principle, an important large deviation result not covered here: Sec. 2.5 of [4], Sec. 4.2.1 of [14], and Sec. II.4 of [15].

## 4 Large deviations of Markov chains

The results of the previous sections would not form much of a theory if they were only applicable to sample means of IID RVs. The power and beauty of large deviation theory is that it can be applied to study the distribution of many different types of quantities or observables involving correlated RVs. This is illustrated by the examples listed in the first section, as well as the lectures given at the school on systems as diverse as interacting particles driven in nonequilibrium states, turbulent fluids, and random matrices, to mention only those.

In this section, we move to the subject of large deviations of correlated systems by considering the simplest case or application, namely, that of sample means of RVs coming from a Markov chain. As will be seen, the computation of a rate function still follows in this case from the SCGF, but the latter is obtained now from an eigenvalue computation that results from the multiplicative form of the generating function. To simplify the presentation, we derive this result in full for Markov chains evolving in discrete time, show some applications for this case, and then state the generalisation of the results for Markov chains evolving in continuous time. The case of stochastic differential equations, defining Markov processes evolving in both continuous space and time, is discussed in the next section.

### 4.1 Markov chains in discrete time

We consider, as in the previous section, a sample mean $S_n$ of $n$ RVs $X_1, \ldots, X_n$, but now assume that the RVs are correlated according to a Markov chain instead of being IIDs. This means that instead of having

$$P(X_1, \ldots, X_n) = P(X_1)P(X_2) \cdots P(X_n), \tag{84}$$

for the joint probability of the sequence, $P$ being the sampling distribution of the $X_i$'s, we assume that

$$P(X_1, \ldots, X_n) = P(X_1)P(X_2|X_1) \cdots P(X_n|X_{n-1}), \tag{85}$$

where $P(X_n|X_{n-1})$ is the conditional probability of $X_n$ given $X_{n-1}$, describing the correlation between these two RVs. Note that, in general, a joint distribution factorises as

$$P(X_1, \ldots, X_n) = P(X_1)P(X_2|X_1)P(X_3|X_1, X_2) \cdots P(X_n|X_1, \ldots, X_{n-1}), \tag{86}$$

so the assumption underlying a Markov chain, as a probabilistic model, is that

$$P(X_n|X_1, \ldots, X_{n-1}) = P(X_n|X_{n-1}). \tag{87}$$

By seeing the sequence $X_1, X_2, \ldots, X_n$ as an ordered sequence in time, we then say that a Markov chain is such that the future state $X_n$ only depends on the present state $X_{n-1}$ and not

the past states coming before $X_{n-1}$. This is often expressed by representing the sequence of RVs in the following way:

$$X_1 \rightarrow X_2 \rightarrow \cdots \rightarrow X_n \,, \tag{88}$$

to emphasise that $X_2$ depends on $X_1$, $X_3$ on $X_2$ and so on.

We give in the Further reading section some references on the theory of Markov chains and its applications in statistical physics. For our purpose, we note that the conditional probability $P(X_n|X_{n-1})$ defines a table or matrix giving the probability of going from a state $X_{n-1}$ at time $n-1$ to any state $X_n$ at the next time $n$. This matrix is called the *transition matrix* and will be denoted by

$$\Pi_{ij} = P(X_n = j | X_{n-1} = i) \,. \tag{89}$$

In this form, we follow the convention used in mathematics of representing the starting state $i$ as the row index and the target (transition-to) state $j$ as the column index. In physics, the reverse convention is used, so that

$$\Pi'_{ij} = P(X_n = i | X_{n-1} = j) \,, \tag{90}$$

and thus $\Pi = (\Pi')^{\mathsf{T}}$. Here, we use $\Pi$, not $\Pi'$, and assume for simplicity that $\Pi$ does not depend explicitly on the time $n$, so the Markov chains that we consider are homogeneous.

The transition matrix is a stochastic matrix satisfying the following properties:

- $0 \leq \Pi_{ij} \leq 1$ for all states $i, j$ (assumed discrete).

- $\sum_j \Pi_{ij} = 1$ for all $i$.

By marginalising the joint distribution in (85) down to the marginal distribution $P(X_n)$, it is also easy to see that the transition matrix is responsible for the evolution of probabilities in time, meaning that

$$P(X_n = j) = \sum_i P(X_{n-1} = i) \Pi_{ij} \,. \tag{91}$$

This is a matrix equation known as the *Chapman–Kolmogorov equation*. Let $\pi(n)$ be a row vector with components $\pi(n)_j = P(X_n = j)$. Then the above equation becomes

$$\pi(n) = \pi(n-1)\Pi \,, \tag{92}$$

so the probabilities of occupying the different states at time $n$ is obtained by multiplying the probability vector at the previous time with the matrix $\Pi$. By iterating from the initial probability vector $P(X_1)$, we then have

$$\pi(n) = \pi(1)\Pi^{n-1} \,, \tag{93}$$

$\Pi^{n-1}$ being the $(n-1)$th matrix power of $\Pi$. This shows that $\pi(n)$ depends in general on the different transition probabilities specifying the Markov chain and the choice of initial probabilities over the states.

In many of the exercises that we will consider, the Markov chain specified by $\Pi$ is *ergodic*, in the sense that there exists a unique probability vector $\pi^*$ such that

$$\pi^* = \lim_{n \to \infty} \pi(n) \,, \tag{94}$$

starting from any initial distribution $\pi(1)$. This means that the Markov chain reaches a statistical steady state characterised by time-invariant occupation probabilities for the different states. As such, $\pi^*$ must then satisfy

$$\pi^* = \pi^*\Pi \,, \tag{95}$$

so it is a row eigenvector of $\Pi$ with eigenvalue 1.

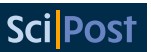

Figure 7: Graphical representation of the asymmetric binary Markov chain.

Ergodic Markov chains are important in statistical physics and other applications, such as numerical sampling (see Further reading). However, it is important to keep in mind that not all Markov chains are ergodic, as we show with the next example. Moreover, for applying large deviation theory to Markov chains, we do not need to assume that $\Pi$ is ergodic, as explained in [2].

**Example 8.** Like the Bernoulli RV, the simplest Markov chain we can consider is one with two states that we can denote or label by 0 and 1, so that $X_i \in \{0, 1\}$. For this Markov chain, $\Pi$ is a $2 \times 2$ matrix containing the following entries:

$$\Pi = \begin{pmatrix} \Pi_{00} & \Pi_{01} \\ \Pi_{10} & \Pi_{11} \end{pmatrix} = \begin{pmatrix} 1-\alpha & \alpha \\ \beta & 1-\beta \end{pmatrix}, \tag{96}$$

where $\alpha \in [0, 1]$ and $\beta \in [0, 1]$ (see Fig. 7). Depending on the value of these two parameters, the Markov chain is ergodic or not:

- For $\alpha$ and $\beta$ in $(0, 1)$, the Markov chain is ergodic with the following stationary distribution:

$$\pi^* = \begin{pmatrix} \frac{\beta}{\alpha+\beta} & \frac{\alpha}{\alpha+\beta} \end{pmatrix}. \tag{97}$$

- For $\alpha = 1$ and $\beta = 1$, the Markov chain is not ergodic. Starting with $\pi(1) = \begin{pmatrix} a & 1-a \end{pmatrix}$, we get $\pi(2) = \begin{pmatrix} 1-a & a \end{pmatrix}$, and then $\pi(3) = \pi(1)$, so the Markov chain is periodic with a period 2.

- For $\alpha = 0$ and $\beta = 0$, the Markov chain is trivial: $\Pi$ is the identity matrix, so there is no evolution of the initial probability vector. As a result, any $\pi(1)$ is an invariant distribution, that is, $\pi(n) = \pi(1)$ for all $n$, which means that there is an infinite number of such distributions, which do not act as attractors for the Chapman–Kolmogorov equation.

## 4.2 Large deviations

We now study the large deviations of sample means of RVs generated by a Markov chain. The sample mean is our observable, defined as before as

$$S_n = \frac{1}{n} \sum_{i=1}^{n} X_i. \tag{98}$$

What changes is our underlying (microscopic) model for the $X_i$'s, which are now linked by a Markov chain with transition matrix $\Pi$, assumed to be ergodic.

As before, we proceed to check that $S_n$ satisfies the LDP and find its rate function $I(s)$ using the Gärtner–Ellis theorem. To this end, we compute the SCGF $\lambda(k)$, considering first the

generating function

$$
\begin{aligned}
E[e^{nkS_n}] &= \sum_{x_1,\dots,x_n} P(x_1,\dots,x_n) e^{k\sum_{i=1}^n x_i} \\
&= \sum_{x_1,\dots,x_n} P(x_1)\Pi_{x_1,x_2}\cdots\Pi_{x_{n-1},x_n} e^{k\sum_{i=1}^n x_i} \\
&= \sum_{x_1,\dots,x_n} P(x_1)e^{kx_1}\Pi_{x_1,x_2}e^{kx_2}\cdots\Pi_{x_{n-1},x_n}e^{kx_n} ,
\end{aligned}
\tag{99}
$$

having used the Markov property in the second line and distributed the exponential in the third. The last sum represents a repeated matrix product. To make this more obvious, define

$$
\hat{P}_k(x) = P(x)e^{kx} ,
\tag{100}
$$

and

$$
(\hat{\Pi}_k)_{ij} = \Pi_{ij} e^{kj} .
\tag{101}
$$

The modified initial distribution is called the *tilted distribution* (unnormalised), while the modified transition matrix is called the *tilted transition matrix*. Using both, we have

$$
E[e^{nkS_n}] = \sum_{x_n}\sum_{x_{n-1}}\cdots\sum_{x_2}\left(\sum_{x_1}\hat{P}_k(x_1)(\hat{\Pi}_k)_{x_1,x_2}\right)(\hat{\Pi}_k)_{x_2,x_3}\cdots(\hat{\Pi}_k)_{x_{n-1},x_n} .
\tag{102}
$$

The first sum between the parentheses represents a product of the row vector $\hat{P}_k$ and the matrix $\hat{\Pi}_k$, whose result (a row vector) is multiplied again with $\hat{\Pi}_k$ multiple times up to the sum involving $x_{n-1}$. The last sum over $x_n$ adds the components of the last row vector obtained, so we can rewrite the generating function in the end as

$$
E[e^{nkS_n}] = \sum_{x_n}(\hat{P}_k\hat{\Pi}_k^{n-1})_{x_n} .
\tag{103}
$$

This can be interpreted as the trace of a vector, though this is not important. More crucial is the fact that we are dealing with a repeated product of a positive matrix $\hat{\Pi}_k$, starting with an arbitrary vector $\hat{P}_k$, which is also positive. As a result, we can invoke the Perron–Frobenius theorem to say that, as $n \to \infty$, the repeated product becomes dominated by the largest eigenvalue of $\hat{\Pi}_k$. To be more precise, let us denote the eigenvalues of $\hat{\Pi}_k$ as $\zeta_{k,i}$ and the corresponding row (left) and column (right) eigenvectors as $l_{k,i}$ and $r_{k,i}$, respectively. Note that $\hat{\Pi}_k$ has in general different sets of left and right eigenvectors because it is not necessarily symmetric. Expanding the sum of the generating function in the (bi-orthogonal) basis formed by these eigenvectors, we obtain

$$
E[e^{nkS_n}] = \sum_{x_n}\sum_{i} a_{k,i}\zeta_{k,i}^{n-1}(l_{k,i})_{x_n} ,
\tag{104}
$$

where $a_{k,i} = \hat{P}_k r_{k,i}$ is the projection of $\hat{P}_k$ onto $r_{k,i}$ (row vector times column vector). This is perhaps more obvious for physicists if we express the eigen-decomposition of $\hat{\Pi}_k$ in quantum notations as

$$
\hat{\Pi}_k = \sum_{i} \zeta_{k,i} |r_{k,i}\rangle\langle l_{k,i}| .
\tag{105}
$$

Then

$$
E[e^{nkS_n}] = \sum_{x_n}(\hat{P}_k\hat{\Pi}_k^{n-1})_{x_n} = \sum_{i}\langle \hat{P}_k|r_{k,i}\rangle\zeta_{k,i}^{n-1}\mathrm{Tr}\langle l_{k,i}| ,
\tag{106}
$$

the trace in the last expression representing the sum of the components of the row or bra vector $\langle l_{k,i}|$.

It is at this point that we use the Perron–Frobenius theorem: the eigenvalue with largest real part is real, since $\hat{\Pi}_k$ is a positive matrix, so the sum above is dominated by this eigenvalue, resulting in

$$E[e^{nkS_n}] \sim \zeta_{\max}(\hat{\Pi}_k)^n, \tag{107}$$

up to a multiplicative constant, where $\zeta_{\max}(\hat{\Pi}_k)$ is the eigenvalue of $\hat{\Pi}_k$ with largest real part. Taking the limit (42) defining the SCGF, we then finally obtain

$$\lambda(k) = \ln \zeta_{\max}(\hat{\Pi}_k). \tag{108}$$

This is our main result showing, as announced, that the SCGF is obtained from an eigenvalue for Markov chains rather than from a simple generating function, as was the case for IID RVs.

**Example 9.** We calculated the rate function of a sample mean of IID Bernoulli RVs in Example 4. Let us see how this rate function changes when the RVs are correlated according to the Markov chain seen in Example 8. For simplicity, we consider the symmetric case where $\alpha = \beta$, so the transition matrix is

$$\Pi = \begin{pmatrix} 1-\alpha & \alpha \\ \alpha & 1-\alpha \end{pmatrix}. \tag{109}$$

The tilted matrix associated with this Markov chain and the sample mean is, from (101),

$$\hat{\Pi}_k = \begin{pmatrix} 1-\alpha & \alpha e^k \\ \alpha & (1-\alpha)e^k \end{pmatrix}. \tag{110}$$

We leave it as an exercise to compute the dominant eigenvalue of this matrix and to compute the Legendre transform of the corresponding SCGF to obtain the rate function. The SCGF can be found explicitly, but not the Legendre transform, so we have to rely on the parametric form of that transform, discussed in the previous section, to get the rate function. The result is shown in Fig. 8. We note that

- $I(s)$ is centered at $s^* = 1/2$ for all $\alpha \in (0,1)$. This is because the stationary distribution is $p^* = (1/2 \; 1/2)$ regardless of $\alpha$, so the typical value of $S_n$ is always $1/2$.

- The rate function $I(s)$ for $\alpha = 1/2$ corresponds to the rate function calculated for the IID case because the Markov chain is uncorrelated for that value of $\alpha$. (Explain why.)

- For $\alpha < 1/2$, 0s are more likely to be followed by 0s (and similarly for 1s), resulting in long sequences of identical bits (persistence). This leads to more fluctuations at the level of $S_n$, as sequences of bits with unequal number of 0s and 1s are more likely to arise.

- For $\alpha > 1/2$, 0s are more likely to be followed by 1s (and similarly for 1s), resulting in long sequences of alternating bits (anti-persistence) for which $S_n$ fluctuates less.

The sample mean is not the only quantity that we can treat in the context of Markov chains. More generally, we can consider

$$S_n = \frac{1}{n} \sum_{i=1}^{n} f(X_i), \tag{111}$$

where $f$ is any function of the $X_i$'s. The tilted transition matrix then becomes

$$(\hat{\Pi}_k)_{ij} = \Pi_{ij} e^{kf(j)}. \tag{112}$$

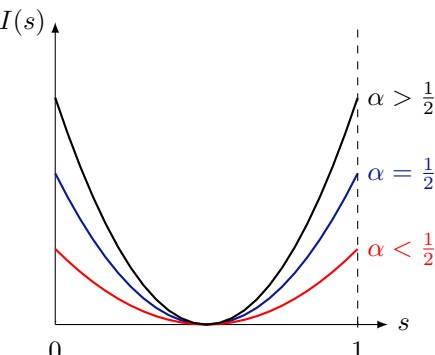

Figure 8: Rate function of the sample mean associated with the symmetric Bernoulli Markov chain for different values of $\alpha$.

Instead of considering a function of $X_i$ at each specific time, we can also consider a function $g(X_i, X_{i+1})$ involving two contiguous RVs in time, so as to define $S_n$ as

$$S_n = \frac{1}{n-1} \sum_{i=1}^{n-1} g(X_i, X_{i+1}). \tag{113}$$

In this case, it can be checked that the calculation leading to the eigenvalue form of the SCGF in (108) still carries through with the tilted matrix

$$(\hat{\Pi}_k)_{ij} = \Pi_{ij} e^{k g(i,j)}. \tag{114}$$

We give some exercises at the end of this section related to this "two-point" observable, which plays an important role in physics. Following this more general observable, it is tempting to look at other observables of the form

$$S_n = \frac{1}{n-2} \sum_{i=1}^{n-2} h(X_i, X_{i+1}, X_{i+2}), \tag{115}$$

or even

$$S_n = \frac{1}{n-m} \sum_{i=1}^{n-m} h(X_i, \ldots, X_{i+m}). \tag{116}$$

Unfortunately, for those the SCGF is not associated with a tilted matrix: the generating function loses its multiplicative form, so the Perron–Frobenius theorem cannot be used to express the SCGF in terms of an eigenvalue, as can be checked explicitly by re-doing the calculations leading to (108).

## 4.3 Sanov's theorem for Markov chains

We have seen in Sec. 3.2 on Sanov's theorem that the empirical frequencies

$$L_n(x) = \frac{1}{n} \sum_{i=1}^{n} \delta_{X_i, x} \tag{117}$$

of the values or states appearing in an IID sequence of RVs follows the LDP with a rate function given by the relative entropy. How is this result changing for a Markov chain?

If the Markov chain is ergodic, then $L_n(x)$ converges to the stationary distribution $p^*$ as $n \to \infty$, similarly to the IID case. This convergence is expressed mathematically by the ergodic

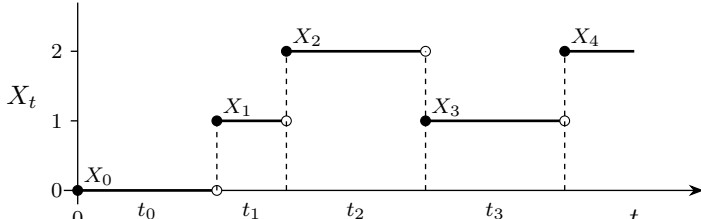

Figure 9: Trajectory of a continuous-time Markov chain $X_t$ over a discrete set of states.

theorem, which is the Markov generalisation of the law of large numbers. Moreover, as in the IID case, $L_n$ concentrates on $p^*$ exponentially with $n$, and so follows the LDP

$$P(L_n = l) \asymp e^{-nI(l)}. \tag{118}$$

What changes is the form of the rate function, which is now given by

$$I(l) = \max_{u>0} \sum_i l_i \ln \frac{u_i}{(u\Pi)_i}, \tag{119}$$

where $u$ is a vector having the same dimension as $l$, so the maximisation is over all vectors with strictly positive entries. Moreover, $u\Pi$ is the product of $u$, taken as a row vector, with the matrix $\Pi$. We leave it as an exercise to check that this rate function is positive, convex, and has a single minimum and zero, located at $l^* = p^*$, confirming that $L_n$ converges exponentially to the stationary distribution satisfying $p^*\Pi = p^*$.

This result was derived by Donsker and Varadhan, who developed the theory of large deviations for Markov processes in a series of mathematical papers in the 1970s. Since then, easier derivations have appeared, including ones based on the Gärtner–Ellis theorem (see the exercises and references at the end of the section). In this context, the tilted matrix is obtained by noting that $f(X_i) = \delta_{X_i,x}$, following the more general sample mean in (111), and that $k$ is a vector, as seen in Sec. 3.2, so that

$$(\hat{\Pi}_k)_{ij} = \Pi_{ij} e^{\sum_a k_a \delta_{j,a}} = \Pi_{ij} e^{k_j}. \tag{120}$$

The derivation of the rate function above from this tilted matrix is also left as an exercise.

## 4.4 Markov chains in continuous time

We close this section by showing how the results obtained for Markov chains evolving in discrete time are modified when considering Markov chains evolving in continuous time. Thus, now the trajectory of the process is written as $(X_t)_{t=0}^T$ with time $t$ running continuously from $t = 0$ to a final or horizon time $T$. For simplicity, we still assume that the state is discrete, so the process jumps between states at random times, as shown in Fig. 9. The dynamics of this jump process is described by a set of *transition rates* between states, given by

$$W_{ij} = \lim_{\Delta t \to 0} \frac{P(X_{t+\Delta t} = j | X_t = i)}{\Delta t}. \tag{121}$$

The transition probability $\Pi_{ij}$ that we considered before is thus replaced in continuous time by a transition rate $W_{ij}$, corresponding to the probability per unit time for transitioning from state $i$ to state $j$. Related to this transition rate is the *escape rate*

$$r_i = \sum_{j \neq i} W_{ij}, \tag{122}$$

giving the total rate for leaving the state $i$. In terms of both $W_{ij}$ and $r_i$, the *generator* of the Markov jump process is expressed as

$$L_{ij} = W_{ij} - r_i \delta_{ij}. \tag{123}$$

This is the matrix determining, similarly to the Chapman–Kolmogorov equation, the evolution of the probability vector $\pi_i(t) = P(X_t = i)$ as

$$\frac{d}{dt}\pi(t) = \pi(t)L. \tag{124}$$

As in (92), this is a vector equation involving the row vector $\pi(t)$ multiplied with the (square) matrix $L$, containing the elements $W_{ij}$ as its off-diagonal elements and $r_i$ in its diagonal.

Markov jump processes are used extensively in statistical physics to model stochastic processes with a discrete state, such as lasers, heat engines, and chemical reactions, for example (see the references at the end of the section for more). Our interest in these systems is in calculating the distribution of observables, defined in a general way now as

$$S_T = \frac{1}{T}\int_0^T f(X_t)dt + \frac{1}{T}\sum_{t\in[0,T]:X_{t^-}\neq X_{t^+}} g(X_{t^-},X_{t^+}). \tag{125}$$

The integral generalises the sample mean that we considered before for IID RVs and Markov chains, whereas the sum follows the generalisation of the sample mean that we discussed for Markov chains, involving the two-point function $g(X_{t^-},X_{t^+})$. Here, the sum is over all the jumps of the process for which the state $X_{t^-}$ before a jump occurs at a time $t$ is different from the state $X_{t^+}$ right after the jump. The actual forms of $f$ and $g$ depend on the application considered, as illustrated with the exercises at the end of the section.

Similar to the IID case, finding the distribution of $S_T$ is difficult, if not impossible, so we look instead at approximating this distribution using the LDP

$$P(S_T = s) \asymp e^{-TI(s)}, \tag{126}$$

which now holds in the long-time $T \to \infty$. To find the rate function $I(s)$, we can still use the Gärtner–Ellis theorem, taking care to modify the SCGF to

$$\lambda(k) = \lim_{T\to\infty} \frac{1}{T}\ln E\left[e^{TkS_T}\right]. \tag{127}$$

As for Markov chains, it can be shown that the generating function $E\left[e^{TkS_T}\right]$ underlying the SCGF has a multiplicative structure (see the next section), which can be used to express the SCGF in terms of a dominant eigenvalue of some matrix. We leave the derivation of this result as an exercise (see the next section for directions) and only mention the end result, namely,

$$\lambda(k) = \zeta_{\max}(\mathcal{L}_k), \tag{128}$$

where

$$(\mathcal{L}_k)_{ij} = W_{ij}e^{kg(i,j)} - r_i\delta_{ij} + kf(i)\delta_{ij}. \tag{129}$$

The matrix above is called the *tilted generator*. The reason the SCGF is now given by an eigenvalue and not by the log of an eigenvalue is essentially that the transition probability matrix $\Pi_{\Delta t}$ describing the dynamics of a jump process over an infinitesimal time interval $\Delta t$ is given in terms of the generator $L$ by $\Pi_{\Delta t} = e^{\Delta t L}$, so that the logarithm of the dominant eigenvalue of the tilted version of $\Pi_{\Delta t}$ corresponds to the dominant eigenvalue of the tilted version of $L$ (after properly accounting for the time limit).

## 4.5 Exercises

- Large deviations of Markov chains: Exercises 3.6.10-3.6.13 of [4].

- Show that $\lambda(0) = 0$ from the eigenvalue representation of the SCGF for both discrete-time and continuous-time Markov chains.

- Markov generalisation of Sanov's theorem: Derive the form of the rate function in (119) for the empirical vector of a Markov chain starting from the Gärtner–Ellis theorem. Obtain an equivalent result for continuous-time Markov chains. From the results, check that the rate function has a unique minimum corresponding to the stationary distribution.

- Derive the dominant eigenvalue representation of the SCGF for a continuous-time Markov chain by discretising the dynamics of the Markov chain in time. Source: Sec. 3.3 of [4].

## 4.6 Further reading

- Theory of Markov chains (discrete and continuous time): Chap. 6 of [19]; Chap. 8 of [20] for jump processes.

- Large deviations of Markov chains: Sec. 3.1 of [14]; Chap. IV of [15]; Sec. 4.3 of [2].

- Sanov's theorem for Markov chains: Sec. 3.1.2 of [14].

- Large deviations of physical jump processes and quantum systems: [24].

- Original articles of Donsker and Varadhan on the large deviations of Markov processes (very technical): [25–27].

- Applications of Markov chains and jump processes: [20], [21], [22], [23].

- Use of Markov chains in sampling: [28].

# 5 Large deviations of Markov diffusions

We continue in this section with the large deviations of Markov processes by considering the case of continuous-state and continuous-time Markov processes defined by stochastic differential equations (SDEs). This is an important class of processes used in physics to model systems such as diffusing Brownian particles subjected to various forces, the transport of particles and energy at macroscopic scales, and the evolution of dynamical systems perturbed by external noise, among many other examples (see Further reading). The study of large deviations in this case also involves the computation of a dominant eigenvalue, but there is an important difference compared with the discrete-state jump processes considered in the previous section: instead of having to compute the dominant eigenvalue of a matrix (either $\Pi_k$ or $L_k$), we now have to compute this eigenvalue for a linear differential operator, corresponding to a tilted version of the generator of the SDE considered. As in the previous section, we explain this result without much derivation, especially since it is well explained in a different set of lecture notes [5]. The exercises suggested at the end of the section are also taken from those lecture notes.

## 5.1 Stochastic differential equations

An SDE is a noisy differential equation determining the evolution of a state $X_t$ in time, defined in mathematics as a difference equation of the form

$$dX_t = F(X_t)dt + \sigma dW_t, \tag{130}$$

where

- $X_t \in \mathbb{R}^n$ is the state at time $t$.

- $dX_t = X_{t+dt} - X_t$ is the increment of the state over the infinitesimal time interval $dt$.

- $W_t \in \mathbb{R}^m$ is a vector of independent Brownian motions modelling the noise disturbing $X_t$, characterised by $E[dW_t]$ and $E[dW_t dW_{t'}] = 0$ for $t' \neq t$ and $E[dW_t^2] = dt$.

- $F : \mathbb{R}^n \to \mathbb{R}^n$ is the force or *drift* driving $X_t$ in a deterministic way.

- $\sigma$ is an $n \times m$ matrix, called the noise matrix.

To simplify the presentation, we consider $\sigma$ to be a constant matrix (more generally, it could depend on $X_t$) and also consider the case where $W_t$ has the same number of components as $X_t$, i.e., $n = m$, so that $\sigma$ is a square matrix.

In physics, the same model is more usually defined as a noisy differential equation, written as

$$\dot{X}_t = F(X_t) + \sigma \xi_t, \tag{131}$$

where $\dot{X}_t = dX_t/dt$ is the time derivative, so that $\xi_t = dW_t/dt$. In this form, $\xi_t$ is called a *Gaussian white noise* and is such that $E[\xi_t] = 0$ and

$$E[\xi_t \xi_{t'}] = \delta(t - t'). \tag{132}$$

The singular form of this covariance involving a Dirac delta function arises because $W_t$ is not differentiable in time and explains why mathematicians prefer to write an SDE as a difference equation, involving the non-singular noise increments $dW_t$ as above, and not as a differential equation.

The form of the drift $F$ and the noise matrix $\sigma$ is determined by the physical system one is trying to model with the SDE. Once these are specified, we can start making predictions about the evolution of $X_t$ via its probability density $p(x, t) = p(X_t = x)$, which satisfies the *Fokker–Planck equation*:

$$\frac{\partial}{\partial t} p(x, t) = -\nabla \cdot \left( F(x) p(x, t) \right) + \frac{1}{2} \nabla \cdot D \nabla p(x, t), \tag{133}$$

starting from an initial density $p(x, 0)$. Here, $\nabla$ is the usual (vector) gradient operator, $\cdot$ is the scalar product, and $D = \sigma \sigma^\mathsf{T}$ is the *diffusion matrix,* which is symmetric by definition. For the remainder, it is useful to note that this partial differential equation (PDE) for $p(x, t)$, which is the continuous-time and continuous-space analog of the Chapman–Kolmogorov equation, is linear in space, so we can put it in the form

$$\frac{\partial}{\partial t} p(x, t) = L^\dagger p(x, t), \tag{134}$$

defining

$$L^\dagger = -\nabla \cdot F + \frac{1}{2} \nabla \cdot D \nabla, \tag{135}$$

as the *Fokker–Planck operator* or *generator*.

Though simple in form, the Fokker–Planck equation cannot be solved easily in general, except for simple SDEs, such as one-dimensional or linear SDEs. From the solution $p(x,t)$, we can determine the probability that $X_t$ reaches any set of states in time by integration:

$$P(X_t \in A) = \int_A p(x,t)dx. \tag{136}$$

In addition, we can find the evolution of expectations of $X_t$, defined in a general way as

$$E[\phi(X_t)] = \int_{\mathbb{R}^n} \phi(x)p(x,t)\,dx, \tag{137}$$

where $\phi : \mathbb{R}^n \to \mathbb{R}$ is any real (and integrable) function of $X_t$. These expectations also evolve in time according to a linear PDE, given by

$$\frac{\partial}{\partial t}E[\phi(X_t)] = E[(L\phi)(X_t)], \tag{138}$$

where

$$L = F \cdot \nabla + \frac{1}{2}\nabla \cdot D\nabla. \tag{139}$$

We use $(L\phi)$ in this equation to mean that $L$ is explicitly acting on the function $\phi$, so the evolution of $E[\phi(X_t)]$ follows by first applying $L$ on $\phi$ and by then taking the expectation with respect to $X_t$.

The equation for expectations is more often considered in mathematics, where it is taken to define the operator $L$, called the *Markov generator* of $X_t$, whereas physicists prefer to work with the Fokker–Planck equation and thus with $L^\dagger$. As the notation suggests, $L$ and $L^\dagger$ are adjoints of one another as operators, in a way similar to the notion of adjoint in quantum mechanics. This is explained in detail in Appendix A of [5]. One essential difference with quantum mechanics is that $L$ is not self-adjoint, in general, because of the minus sign difference between $L$ and $L^\dagger$ coming in front of $F$. This will come to play an important role when we get to large deviations.

## 5.2 Observables and LDP

As before, we are not interested so much in the probability distribution of $X_t$ as in the distribution of observables associated with that state, defined as time-integrals of functions of $X_t$. For physical applications, the general class of observables that needs to be considered has the form

$$A_T = \frac{1}{T}\int_0^T f(X_t)\,dt + \frac{1}{T}\int_0^T g(X_t) \circ dX_t, \tag{140}$$

where $f : \mathbb{R}^n \to \mathbb{R}$ is a real function of the state and $g : \mathbb{R}^n \to \mathbb{R}^n$ is a vector field in $\mathbb{R}^n$ which is multiplied in a scalar product way with the increment $dX_t$, so $g(X_t) \circ dX_t$ is a scalar. In this context, we use the symbol $\circ$ rather than $\cdot$ to indicate that the scalar product is calculated using the Stratonovich or midpoint convention in the stochastic integral. Other conventions, such as the Itô convention, can be used, but the Stratonovich convention is generally preferred in physics because it follows the normal rules of calculus and is physically consistent as a result. The observables listed next give some idea about the kind of functions $f$ and $g$ that appear in applications.

- **Empirical distribution:**

$$\rho_T(x) = \frac{1}{T}\int_0^T \delta(X_t - x)\,dt. \tag{141}$$

This gives the fraction of time that $X_t$ spends at the point $x$ in the time interval $[0, T]$. From this density, we can also obtain the fraction of time that $X_t$ spends in a set $A \subseteq \mathbb{R}^n$ by integration:

$$\rho_T(A) = \int_A \rho_T(x)\,dx = \frac{1}{T}\int_0^T \mathbb{1}_A(X_t)\,dt\,, \tag{142}$$

$\mathbb{1}_A(x)$ being the indicator function equal to 1 when $x \in A$ and 0 otherwise. It is important to note that $\rho_T(x)$ and $\rho_T(A)$ are not probabilities; in particular, $\rho_T(x)$ is not the same as $p(x, t)$.

- **Empirical current:**

$$J_{T,i}(x) = \frac{1}{T}\int_0^T \delta(X_t - x)\mathbf{e}_i \circ dX_t\,, \tag{143}$$

where $\mathbf{e}_i$ is the $i$th unit vector in the canonical basis of $\mathbb{R}^N$. Putting all the components in a vector $J_T(x)$ gives us a vector field observable, interpreted as the mean velocity of $X_t$ at the point $x$. To make this more explicit, we could rewrite the integral above as

$$J_T(x) = \frac{1}{T}\int_0^T \delta(X_t - x)\dot{X}_t\,dt\,, \tag{144}$$

but the problem is that, as for $W_t$, $X_t$ has no time derivative.

- **Potential energy difference:** If $X_t$ represents some physical system for which there is an energy function $U(x)$, then the change of energy over time is given by

$$\Delta U_T = U(X_T) - U(X_0) = \int_0^T \nabla U(X_t) \circ dX_t\,. \tag{145}$$

The use of the Stratonovich (scalar) product in the stochastic integral is important. With other conventions (e.g., Itô), other unphysical terms would appear in addition to $\Delta U(X_t)$.

- **Mechanical work:** If the drift $F$ in the SDE represents some kind of force, then the mean work done by this force over time, corresponding to the expended power, is

$$W_T = \frac{1}{T}\int_0^T F(X_t) \circ dX_t\,. \tag{146}$$

The Stratonovich convention must also be used here on physical grounds.

Many other observables can be defined, depending on the application considered, leading to different choices of functions $f$ and $g$. The potential energy and work are especially important when studying the thermodynamics of small stochastic systems driven by external forces, boundary reservoirs, and noise (see Further reading).

Having defined the evolution of $X_t$ (via $F$ and $\sigma$) and the observable $A_T$ (via $f$ and $g$), we are now ready to determine the distribution $P(A_T = a)$ of $A_T$. As before, this is a difficult problem, in general, so we fall back on approximating this distribution in the long-time limit with the LDP

$$P(A_T = a) \asymp e^{-TI(a)}\,. \tag{147}$$

Our goal is to find the rate function

$$I(a) = \lim_{T \to \infty} -\frac{1}{T}\ln P(A_T = a)\,. \tag{148}$$

This will be done next by calculating the SCGF

$$\lambda(k) = \lim_{T \to \infty} \frac{1}{T} \ln E\left[e^{TkA_T}\right]. \tag{149}$$

We need some assumptions to make sure that these limits exist. The most important is to assume that $X_t$ is ergodic, so that $p(x, t)$ converges to a unique stationary probability density $p^*(x)$ satisfying $L^\dagger p^* = 0$. In this case, $A_T$ has a stationary expectation, given by

$$a^* = \lim_{T \to \infty} E[A_T] = \int_{\mathbb{R}^n} f(x) p^*(x) \, dx + \int_{\mathbb{R}^n} J^*(x) \cdot g(x) \, dx, \tag{150}$$

where

$$J^*(x) = F(x) p^*(x) - \frac{D}{2} \nabla p^*(x), \tag{151}$$

is the *stationary probability current* (a vector field) satisfying $\nabla \cdot J^* = 0$. From the ergodic theorem, we also have that $A_T$ converges to $a^*$ in probability as $T \to \infty$, so $a^*$ is not just the stationary expectation – it is the typical value or concentration point of $A_T$ at which $P(A_T = a)$ concentrates exponentially as $T$ increases. This concentration is important for having the LDP and implies, as we saw before, that $I(a^*) = 0$.

Because of the form of $a^*$ above, it is customary in physics to use the following nomenclature:

- **Density-type observables:** Observables $A_T$ for which $f \neq 0$ and $g = 0$. For those, $a^*$ only involves the stationary density $p^*$.

- **Current-type observables:** Observables $A_T$ that involve $g$ but not $f$, such as the work above. For those, $a^*$ is expressed in terms of the stationary current $J^*$.

Naturally, the empirical density is a density-type observable whose typical value is $p^*$, while the typical value of the empirical current $J_T$ is the stationary current $J^*$ (see Exercises), which explains the name "empirical current". As a result, we have $\rho_T \to p^*$ and $J_T \to J^*$, the limit being interpreted in the sense of convergence in probability as $T \to \infty$, the convergence of the law of large numbers and the ergodic theorem. Because of these two limits, $p^*$ and $J^*$ are often estimated in experiments or simulations involving SDEs via $\rho_T$ and $J_T$.

## 5.3 Calculation of the SCGF

Once again, we use the Gärtner-Ellis theorem to obtain the rate function $I(a)$ as the Legendre transform of the SCGF $\lambda(k)$, obtaining the SCGF itself from an eigenvalue calculation, similar to the case of Markov chains. For SDEs, the eigenvalue connection is derived by considering the generating function of $A_T$ conditioned on the initial point $X_0 = x$, written as

$$G(x, k, t) = E[e^{ktA_t} | X_0 = x]. \tag{152}$$

The evolution of this function in time satisfies another linear PDE, referred to as the *Feynman–Kac formula* or *equation*:

$$\frac{\partial}{\partial t} G(x, k, t) = \mathcal{L}_k G(x, k, t), \tag{153}$$

where

$$\mathcal{L}_k = F \cdot (\nabla + kg) + \frac{1}{2}(\nabla + kg) \cdot D(\nabla + kg) + kf \tag{154}$$

is called the *tilted generator* (see Further reading). Note that the initial condition of the Feynman–Kac equation is $G(x, k, 0) = 1$, the unit (constant) function. Moreover, for $k = 0$, we

have $\mathcal{L}_{k=0} = L$, so $\mathcal{L}_k$ is a perturbation of the Markov generator, as was the case for continuous-time Markov chains (see Eq. (129) of Sec. 4.4). In fact, the evolution of $G(x, k, t)$ for such a Markov chain is ruled by the same Feynman–Kac equation as above, but with the differential operator in (154) replaced by the matrix operator in (129). For this reason, much of the derivation that we are presenting now can be used to obtain the result presented before in (128) without proof.

At this point, we use the fact that the Feynman–Kac equation is linear to expand its solution in the eigenbasis of $\mathcal{L}_k$. This follows what we did for Markov chains (see Sec. 4.2), with the difference that we are now dealing with a linear operator acting on functions on $\mathbb{R}^n$, so the eigenbasis is constructed as follows:

- As for $\Pi_k$, $\mathcal{L}_k$ is not necessarily self-adjoint, so we must expand $G(x, k, t)$ in the bi-orthogonal basis constructed from the eigenfunctions and eigenvalues of $\mathcal{L}_k$, given by

$$\mathcal{L}_k r_{k,i}(x) = \zeta_{k,i} r_{k,i}(x), \tag{155}$$

and the eigenfunctions and eigenvalues of the adjoint $\mathcal{L}_k^\dagger$, satisfying

$$\mathcal{L}_k^\dagger l_{k,i}(x) = \zeta_{k,i} l_{k,i}(x). \tag{156}$$

The eigenfunctions $r_{k,i}$ and $l_{k,i}$ play the role, respectively, of the right and left eigenvectors of $\Pi_k$, which explains why we used the same symbols for those, as well as for the eigenvalues. Taken together, the eigenfunctions form a complete bi-orthogonal basis satisfying

$$\int_{\mathbb{R}^n} l_{k,i}(x) r_{k,j}(x) dx = \delta_{ij}, \tag{157}$$

and

$$\sum_i l_{k,i}(y) r_{k,i}(x) = \delta(x - y). \tag{158}$$

- Boundary conditions on the eigenfunctions must be imposed for solving the two spectral equations (155) and (156). For the SDEs and observables that we consider on $\mathbb{R}^n$, these are a decaying condition for $l_{k,i}(x)$ and the product $r_{k,i}(x) l_{k,i}(x)$, respectively, so that

$$\lim_{|x| \to \infty} l_{k,i}(x) = 0, \qquad \lim_{|x| \to \infty} r_{k,i}(x) l_{k,i}(x) = 0, \tag{159}$$

coupled with the normalisation conditions

$$\int_{\mathbb{R}^n} r_{k,i}(x) l_{k,i}(x) \, dx = 1, \qquad \int_{\mathbb{R}^n} l_{k,i}(x) \, dx = 1. \tag{160}$$

We refer to [29] for a justification of these conditions.

The formal solution of the generating function is

$$G(x, k, t) = (e^{\mathcal{L}_k t} 1)(x), \tag{161}$$

where $e^{\mathcal{L}_k t}$ is the exponential of $\mathcal{L}_k$, acting as an operator on the unit function, the initial value of $G(x, k, t)$. By integrating (158) with respect to $y$ and using the normalisation condition imposed on $l_{k,i}$ in (160), we have

$$1 = \sum_i r_{k,i}(x). \tag{162}$$

The spectral expansion of the generating function is therefore

$$G(x,k,t) = \sum_i e^{\zeta_{k,i} t} r_{k,i}(x),$$

(163)

the exponential of the eigenvalues appearing because $r_{k,i}$ is an eigenfunction of $\mathcal{L}_k$. This parallels what we had in (104) for Markov chains and so we expect, as in that case,

$$G(x,k,t) \asymp e^{\zeta_{\max}(\mathcal{L}_k) t} r_k(x),$$

(164)

as $t \to \infty$, where $\zeta_{\max}(\mathcal{L}_k)$ is the eigenvalue of $\mathcal{L}_k$ with largest real part and $r_k(x)$ is its corresponding eigenfunction. Taking the limit defining the SCGF then yields

$$\lambda(k) = \zeta_{\max}(\mathcal{L}_k).$$

(165)

In this way, we obtain the SCGF as the dominant eigenvalue of $\mathcal{L}_k$, similarly to the result in (128).

There are some steps missing in this derivation that we leave as exercise. To be more complete, mathematically, we should take care of the following points:

- The spectrum of $\mathcal{L}_k$, unlike that of $\Pi_k$, is not necessarily discrete, so we have to be careful with assuming that the spectral representation of $G(x,k,t)$ is dominated by a largest term, as expressed in (164). A sufficient condition for this Laplace property to hold is for the spectrum to have a gap, i.e., for the two eigenvalues of $\mathcal{L}_k$ with largest real parts to be separated. (Explain why.)

- Show that the dominant eigenvalue $\zeta_{\max}(\mathcal{L}_k)$ is real, so there is a consistent identification with the SCGF, which is real by definition. (This follows by the Perron–Frobenius theorem generalised to positive operators.)

- Relate the generating function $E[e^{TkA_T}]$ to $E[e^{TkA_T}|X_0 = x]$ and show (under some conditions to be stated) that the two have the same Laplace asymptotics, so they lead to the same SCGF.

## 5.4 Symmetrisation

The spectral problem giving the SCGF in (165) is not easy to solve in practice because we must solve the spectral equation in (155) in tandem with the adjoint equation in (156) while enforcing the boundary conditions in (159) and (160). Thus, unlike in quantum mechanics, we have to solve not one, but two spectral problems for eigenfunctions that do not have their own individual boundary condition; rather, the boundary conditions are to be applied on $l_k$ and on the product $r_k l_k$. This arises even though the spectral representation of $G(x,k,t)$, as given in (163), involves $r_{k,i}$ but not $l_{k,i}$.

One case of SDEs and observables for which this problem simplifies to a single spectral problem, is that of ergodic and reversible SDEs, for which $J^* = 0$, and density-type observables. For those, it is known that the spectrum of $\mathcal{L}_k$ is real, so this operator can be conjugated to a self-adjoint operator. In other words, $\mathcal{L}_k$ can be transformed to a self-adjoint operator $\mathcal{H}_k$ by a unitary transformation, defined explicitly by

$$\mathcal{H}_k = p^{*\frac{1}{2}} \mathcal{L}_k p^{*-\frac{1}{2}},$$

(166)

where $p^*$ is the stationary distribution of the SDE. This is an operator transformation acting on a function $\psi(x)$ as follows:

$$\mathcal{H}_k \psi(x) = \sqrt{p^*(x)} \left( \mathcal{L}_k \frac{\psi}{\sqrt{p^*}} \right)(x),$$

(167)

so $\mathcal{L}_k$ is acting on $\psi(x)/\sqrt{p^*(x)}$, giving a function of $x$ which is then multiplied by $\sqrt{p^*(x)}$.

This *symmetrisation transformation* is described in more detail in [5]. For our purpose, we note that, if $J^* = 0$ and the observable $A_T$ is such that $g = 0$, then $\mathcal{H}_k$ as defined above is self-adjoint, meaning $\mathcal{H}_k = \mathcal{H}_k^\dagger$. As a result, its eigenvalues, which are the same as those of $\mathcal{L}_k$, are real and are associated with a single set of eigenfunctions $\psi_{k,i}$, given by

$$\mathcal{H}_k \psi_{k,i}(x) = \zeta_{k,i} \psi_{k,i}(x), \tag{168}$$

and satisfying the simple boundary condition

$$\lim_{|x|\to\infty} \psi_{k,i}(x) = 0, \qquad \int_{\mathbb{R}^n} \psi_{k,i}(x)^2 \, dx = 1. \tag{169}$$

In this case, the SCGF is therefore calculated is a relatively simple way as in quantum mechanics by finding the dominant eigenvalue of $\mathcal{H}_k$, interpreted, by changing the sign of $\mathcal{H}_k$, as the lowest energy or ground state of some Hamiltonian.

An important class of SDEs for which this result is applicable is the class of diffusions in $\mathbb{R}^n$ with gradient drift $F = -\nabla U$, where $U : \mathbb{R}^n \to \mathbb{R}$ is some potential function, and with a noise matrix $\sigma = \epsilon \mathbb{1}$ proportional to the identity matrix, so that $\epsilon \in \mathbb{R}$. In this case, the tilted generator is

$$\mathcal{L}_k = F \cdot \nabla + \frac{\epsilon^2}{2} \nabla^2 + k f, \tag{170}$$

for $g = 0$ and is symmetrised with $p^*(x) \propto e^{-2U(x)/\epsilon^2}$ to

$$\mathcal{H}_k = \frac{\epsilon^2 \nabla^2}{2} - V_k, \tag{171}$$

where

$$V_k(x) = \frac{|\nabla U(x)|^2}{2\epsilon^2} - \frac{\nabla^2 U(x)}{2} - k f(x). \tag{172}$$

This result is interesting: up to a minus sign, this is the Hamiltonian of a quantum particle in a potential $V_k(x)$, which is clearly self-adjoint.

**Example 10.** Consider the Ornstein-Uhlenbeck process

$$dX_t = -\gamma X_t dt + \sigma dW_t, \tag{173}$$

in one dimension, so that $X_t \in \mathbb{R}$, with $\gamma, \sigma > 0$. This is a gradient SDE with quadratic potential, $U(x) = -\gamma x^2/2$, and Gaussian stationary distribution

$$p^*(x) = C e^{-\frac{2U(x)}{\sigma^2}} = C e^{-\frac{\gamma x^2}{\sigma^2}}, \tag{174}$$

where $C$ is a normalisation constant.

For the linear observable

$$A_T = \frac{1}{T} \int_0^T X_t \, dt, \tag{175}$$

interpreted as the mean area, the tilted generator is

$$\mathcal{L}_k = -\gamma x \frac{d}{dx} + \frac{\sigma^2}{2} \frac{d^2}{dx^2} + kx. \tag{176}$$

This operator is not self-adjoint, but can be symmetrised into the self-adjoint $\mathcal{H}_k$, corresponding with (171) and (172) to

$$\mathcal{H}_k = \frac{\sigma^2}{2} \frac{d^2}{dx^2} - V_k(x), \qquad V_k(x) = \frac{\gamma^2 x^2}{2\sigma^2} - \frac{\gamma}{2} - kx. \tag{177}$$

We recognise in this result the Hamiltonian of the quantum harmonic oscillator. From the known spectrum of this system, the SCGF is found to be

$$\lambda(k) = \frac{\sigma^2 k^2}{2\gamma^2}, \tag{178}$$

which gives, by Legendre transform, the rate function

$$I(a) = \frac{\gamma^2 a^2}{2\sigma^2}. \tag{179}$$

In this case, the large deviations are rather trivial: the rate function is quadratic, so the fluctuations of $A_T$ are Gaussian, which is expected given that the integral of a Gaussian process is known to be Gaussian-distributed. A more interesting form of rate function is found by considering $A_T$ to be the integral of $X_t^2$, so that $f(x) = x^2$ (see Exercises).

## 5.5 Exercises

The numbered exercises are in [5].

- Show that the Fokker–Planck equation can be put in a conservative form $\partial_t p = -\nabla \cdot J_t$ and find the expression of $J_t$. Comment on the meaning of $J_t$ and find its relationship with $J^*$.

- Typical value of $A_T$: Exercises 7 and 8.

- Feynman–Kac equation: Exercises 9 and 10.

- Symmetrisation: Exercises 12-14.

- Large deviations of SDEs: Exercises 16-24.

- Redo the spectral calculation for the linear SDE in (10) and the linear observable (175), but now derive all the eigenfunctions $\psi_{k,i}$. From the result, obtain $l_{k,i}$ and $r_{k,i}$, and verify that these satisfy the boundary conditions in (160). In particular, verify that the $r_{k,i}$'s do not decay to 0 at infinity.

- Complete the derivation in Sec. 5.3 leading to (165), following the points listed at the end of the section.

## 5.6 Further reading

- SDEs: [20], [31], [32], and [33].

- Stochastic calculus and conventions: Chap. 3 of [31], Chap. 7 of [32], Chap. 4 of [33].

- Stochastic thermodynamics: [34].

- Large deviations for SDEs: [5].

- Large deviations of linear SDEs: [35].

- Large deviations of SDEs with reflective boundaries: [29] and [30].

- Feynman–Kac equation: Sec. 7.7 of [33], Appendix C and Exercise 9 of [5], as well as [36].

- Historical references on the Feynman–Kac equation: [37], [38].

- Markov generators: Appendix A of [5].

- Bi-orthogonal bases: Appendix B of [5].

- Symmetrisation: Sec. 4.3 of [5].

# 6 How do large deviations arise?

So far, we discussed methods for computing the SCGF and the rate function, which give us information about the fluctuations of time-integrated observables of Markov processes, and more precisely about the distribution of these observables on the exponential scale. In this last section, we want to understand in a more precise way how these fluctuations appear by considering the set of trajectories or paths that lead to a specific fluctuation and by determining whether this set can be described as a Markov process. We explain how this idea is related mathematically to the notion of conditioning a Markov process and how a Markov process, called the effective or fluctuation process, can be extracted from the long-time limit of this conditioning. The effective process turns out to involve the eigenvector or eigenfunction $r_k$, so the spectral problem of the previous sections holds more information about the large deviations: the dominant eigenvalue gives the SCGF and rate function, which tell us how likely a fluctuation is, while the dominant eigenfunction gives the effective process, which tells us how that fluctuation is created.

## 6.1 Conditioning problem

Suppose we are able to observe or simulate a Markov process in time. How should we go about determining the distribution and rate function of some observable of that process? To fix the discussion, consider $X_t$ to be a diffusion process evolving according to some SDE and let us denote the observable by $A_T$, as we did before. To find $P(A_T)$, it is natural to observe or simulate many paths of $X_t$ over the time interval $[0, T]$, compute the value of $A_T$ for each, and construct the histogram of the values or "measurements" obtained. If we know or expect that $P(A_T)$ satisfies the LDP, we then extract its rate function by transforming the histogram according to $-\frac{1}{T} \ln P(A_T = a)$, as we did in Fig. 1, repeating this for many increasing values of $T$, so as to hopefully obtain a meaningful (i.e., converged) function $I(a)$.

This is not the most efficient way of estimating the rate function. Since $P(A_T = a)$ decays exponentially with $T$, we need an exponentially large number of paths to populate the different bins of the histogram of $A_T$ and so to see any fluctuation of this observable away from its typical value $a^*$. Indeed, in this experiment, most paths will be such that $A_T = a^*$, as we depict in Fig. 10 with the paths in blue associated with the region of the distribution around the typical value $a^*$. However, if we are patient enough, we are bound to observe "rare" paths of $X_t$ for which $A_T$ takes some value away from $a^*$. These are shown in red in Fig. 10 and are often found to be very different from the "typical" paths leading to $A_T = a^*$, as suggested by the figure.[5]

In mathematical terms, this selection of paths based on the event $A_T = a$ is equivalent to a conditioning of $X_t$, expressed in the path integral formalism by writing

$$P^a[x] \equiv P[x|A_T = a] = \frac{P[A_T = a|x]P[x]}{P(A_T = a)}, \tag{180}$$

---

[5]In practice, a histogram has bins of a certain size, say $\Delta a$, so the events that we consider, namely, $A_T = a^*$ or $A_T = a$, should really be interpreted as $A_T \in [a^*, a^* + \Delta a]$ and $A_T \in [a, a + \Delta a]$, as in Fig. 10. Since the bin size can be made arbitrarily small and does not play a role in the theory, we use $A_T = a^*$ and $A_T = a$ in the text.

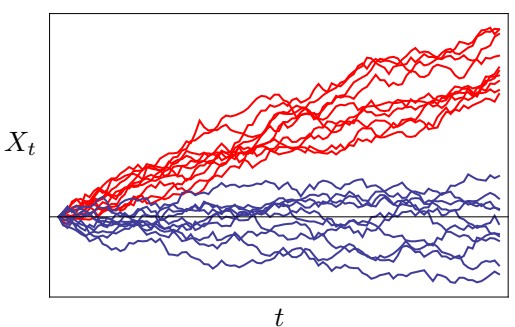 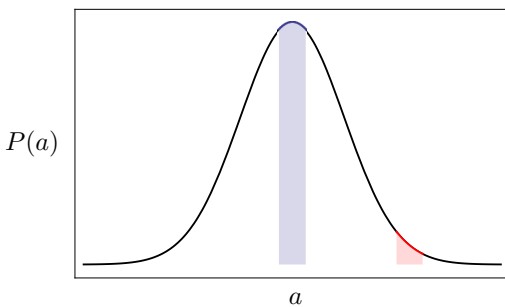

Figure 10: Conditioning a process on large deviations. Paths of a diffusion process (left) leading to different values of an observable $A_T$ with different probabilities $P(a) = P(A_T = a)$ (right). The paths in blue are associated with the typical value $a^*$ of $A_T$, whereas the paths in red are associated with a rare value (viz., fluctuation) of $A_T$ away from $a^*$. The collection of all such paths defines the conditioned process $X_t | A_T = a$.

where $P[x] = P[(x_t)_{t=0}^T]$ is the path probability distribution of $X_t$. This is Bayes's theorem expressed at the level of path probabilities. Noting that $P[A_T = a|x] = \delta(A_T[x] - a)$, we then have

$$P^a[x] = \frac{\delta(A_T[x] - a)}{P(A_T = a)} P[x].$$ (181)

This matches the selection experiment we described: the conditioning of $X_t$ is a stochastic process that reweights the original path distribution onto the paths such that $A_T = a$.

Conceptually, this conditioning is similar to the microcanonical ensemble of statistical mechanics, which assigns the same probability to microstates of an equilibrium system having the same energy. Here, the energy constraint is replaced by the conditioning event $A_T = a$, leading $P^a[x]$ to be non-zero for paths such that $A_T = a$ (consistently with the selection). For this reason, the conditioned process is also called a microcanonical ensemble of paths or dynamical microcanonical ensemble [39]. In the recorded lectures, we denote this process by the notation

$$X_t | A_T = a,$$ (182)

to mean that $X_t$ is conditioned on observing the value $A_T = a$ [40].

At this point, there is no assumption that the conditioned process is a Markov process. In fact, this is the problem that we want to address – to determine whether $X_t | A_T = a$ is Markovian either exactly or approximately.

Already, there should be a sense that we do not have the Markov property exactly for $T < \infty$ because the event $A_T = a$ acts as a global constraint on the whole time interval $[0, T]$. This is perhaps more obvious if we consider the Markov chain version of the conditioned process, corresponding to

$$P(X_1, \ldots, X_n | S_n = s) = \frac{\delta(S_n - s)}{P(S_n = s)} P(X_1) P(X_2 | X_1) \cdots P(X_n | X_{n-1}),$$ (183)

where $S_n$ is the sample mean of the $X_i$'s. In this expression, the Markov structure of the path distribution $P[X] = P(X_1, \ldots, X_n)$ of the Markov chain was used according to (85), but the presence of the reweighting with the delta function and $P(S_n = s)$ prevents us from writing the result as a product of transition matrices. Similarly, if we consider an IID sequence of RVs, then the conditioning gives

$$P(X_1, \ldots, X_n | S_n = s) = \frac{\delta(S_n - s)}{P(S_n = s)} P(X_1) P(X_2) \cdots P(X_n),$$ (184)

which does not have an obvious IID or independent form with a product of distributions.

The same applies to Markov processes evolving in continuous time. That is to say, $X_t|A_T = a$ is not in general a Markov process for a finite observation time $T$, unless the observable is taken to have a trivial form or we consider a different type of conditioning (see Exercises). However, *if we consider the long-time limit $T \to \infty$, then it is known that $X_t|A_T = a$ does become Markovian in an approximate sense* [40]. The full explanation and proof of this result is beyond the scope of these notes (and lectures), so we only define the limiting Markov process and explain how it approximates the conditioned process in the next sections. The basic idea is that it is possible to construct from $X_t$ a new Markov process $\hat{X}_t$, called the *effective*, *driven* or *fluctuation process*, whose path distribution $\hat{P}[x]$ is such that

$$\hat{P}[x] \asymp P^a[x], \tag{185}$$

as $T \to \infty$. Thus, although the conditioned process is not a Markov process in general, it is close to a Markov process on the exponential (asymptotic) scale – the same approximation scale that we use for defining the LDP (see Sec. 2.3). In this sense, the effective process *approximates* the conditioned process in the long-time limit. In particular, it can be shown that $A_T \to a$ if we observe or simulate $\hat{X}_t$ (see Exercises), so the effective process realises the hard constraint $A_T = a$ of the conditioned process in a soft way as a typical value that becomes a concentration point as $T \to \infty$. This is explained in more detail below.

## 6.2 Effective process

The effective process $\hat{X}_t$ is a Markov process described by a generator $L_k$, obtained by transforming the tilted generator $\mathcal{L}_k$ as follows:

$$L_k = r_k^{-1} \mathcal{L}_k r_k - r_k^{-1}(\mathcal{L}_k r_k) = r_k^{-1} \mathcal{L}_k r_k - \lambda(k), \tag{186}$$

where $\lambda(k)$ is the SCGF, corresponding to the dominant eigenvalue of $\mathcal{L}_k$, and $r_k$ is the corresponding eigenfunction. The expression above is an operator transform, called the *generalised Doob transform*, which is similar to the symmetrisation transformation discussed before. As for that transformation, we must understand the expression of $L_k$ above by applying it to a function, obtaining

$$\begin{aligned} L_k \phi(x) &= \frac{1}{r_k(x)}(\mathcal{L}_k r_k \phi)(x) - \frac{1}{r_k(x)}(\mathcal{L}_k r_k)(x)\phi(x) \\ &= \frac{1}{r_k(x)}(\mathcal{L}_k r_k \phi)(x) - \lambda(k)\phi(x). \end{aligned} \tag{187}$$

Thus, the first term $\mathcal{L}_k r_k$ in $L_k$ is not yet effective until we apply it to a function, while the second term $(\mathcal{L}_k r_k)$ with the parentheses indicates that $\mathcal{L}_k$ is first applied on $r_k$ before acting on a function, resulting in the "diagonal" or "constant" term $\lambda(k)\phi(x)$ above.

The theory of the generalised Doob transform and its connection to the conditioned process can be found in [40]. Here, we limit ourselves to list the main results of that theory, namely,

- $L_k$ is a valid Markov generator of a Markov process, satisfying $L_k 1 = 0$ when applied to the unit function, as can be checked from its definition.

- The stationary distribution $p_k^*$ of $\hat{X}_t$ satisfying $L_k^\dagger p_k^* = 0$ is

$$p_k^*(x) = r_k(x)l_k(x), \tag{188}$$

  where $l_k$ is the eigenfunction of $\mathcal{L}_k^\dagger$ associated with $\lambda(k)$ (see Exercises). This probability density is normalised as a result of the normalisation chosen in (160).

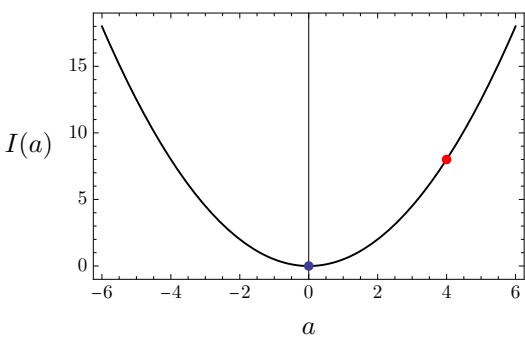
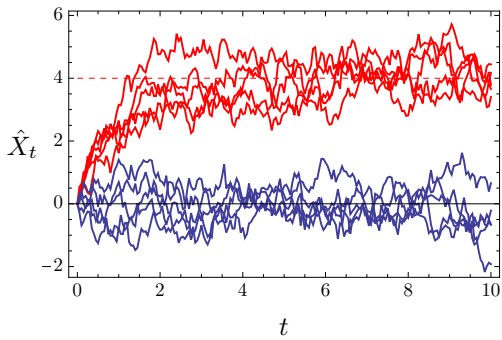

Figure 11: Effective process associated with area fluctuations of the Ornstein–Uhlenbeck process. Blue: Simulated paths of the original process associated with the typical value $a^* = 0$. Red: Simulated paths of the effective process associated with the fluctuation value $A_T = 4$. Parameters: $\gamma = 1$ and $\sigma = 1$.

- Although this is not made explicit by writing $\hat{X}_t$, the effective process depends on $k$, as its generator $L_k$ depends on $k$ via $\mathcal{L}_k$, $r_k$, and $\lambda(k)$. To relate this process to the conditioning constraint $A_T = a$, we must choose $k$ such that

$$\lambda'(k) = a, \qquad \text{or } I'(a) = k. \tag{189}$$

These are the duality relations, seen in Sec. 2.6, linking the SCGF and the rate function.

- Choosing $k$ according to the duality relation above, we have $A_T \to a$ in probability as $T \to \infty$, so $A_T$ concentrates in the effective process to $a$, as mentioned before. If we just select a value of $k$, without relating it to $a$, then we have $A_T \to a^*(k)$, where

$$a^*(k) = \lambda'(k) = \lim_{T \to \infty} E_{p_k^*}[A_T]. \tag{190}$$

This is the asymptotic expectation of $A_T$ with respect to the effective process, which is also, as we know, the typical value and concentration point of $A_T$.

- For $k = 0$, the effective process is the same as the original process. This is consistent with the fact that $A_T \to a^*$ in the original process and that $\lambda'(0) = a^*$.

These results hold for general Markov processes evolving in continuous time, including the discrete jump processes and diffusions considered before. For diffusions, specifically, the effective process can be expressed directly in terms of an SDE that keeps the noise matrix of the original diffusion $X_t$, but changes the drift according to

$$\hat{F}_k(x) = F(x) + D[kg(x) + \nabla \ln r_k(x)]. \tag{191}$$

The SDE of the effective process is thus

$$d\hat{X}_t = \hat{F}_k(\hat{X}_t)dt + \sigma dW_t. \tag{192}$$

This depends again on $k$: to relate $\hat{X}_t$ to the dynamics of the conditioned process, we must fix $k$ according to the duality relations in (189). This is illustrated next with the Ornstein–Uhlenbeck process, studied before in Example 10.

**Example 11.** Consider again the Ornstein–Uhlenbeck process defined by the SDE in (173) with the area observable in (175). We found for this system the quadratic SCGF in (178), which is actually associated with the following eigenfunction (see Exercises):

$$r_k(x) = e^{\frac{kx}{\gamma} - \frac{3\sigma^2 k^2}{4\gamma^3}}. \tag{193}$$

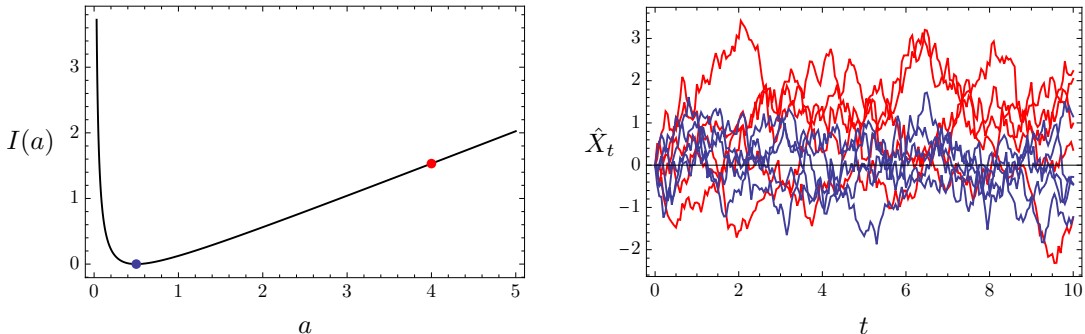

Figure 12: Effective process associated with variance fluctuations of the Ornstein–Uhlenbeck process. Blue: Simulated paths of the original process associated with the typical value $a^* = 1/2$. Red: Simulated paths of the effective process associated with the fluctuation value $A_T = 4$. Parameters: $\gamma = 1$ and $\sigma = 1$.

Plugging this result in (191) then yields with $F(x) = -\gamma x$ and $g(x) = 0$,

$$\hat{F}_k(x) = -\gamma\left(x - \frac{\sigma^2 k}{\gamma^2}\right). \tag{194}$$

This is the effective drift as a function of $k \in \mathbb{R}$. To relate this drift to a fluctuation $A_T = a$, we use again the SCGF to obtain

$$\lambda'(k) = \frac{\sigma^2}{\gamma^2} k, \tag{195}$$

leading to

$$\hat{F}_{k(a)}(x) = -\gamma(x - a). \tag{196}$$

This shows that the paths of the Ornstein–Uhlenbeck process leading to $A_T = a$ have a dynamics that can be described, effectively, as that of another Ornstein–Uhlenbeck re-centred around $a$. In this new process, the states hover around $x = a$ instead of $x = 0$, as illustrated in Fig. 11, leading naturally to the asymptotic mean $A_T \to a$. This can be verified explicitly by computing the stationary distribution of $\hat{X}_t$ to find that it is the same Gaussian density as $X_t$, shown in (174), but re-centered to $a$, so that

$$p^*_{k(a)}(x) = r_{k(a)}(x)l_{k(a)}(x) = p^*(x - a). \tag{197}$$

**Example 12.** If we replace the linear observable in the previous example by the empirical variance

$$A_T = \frac{1}{T}\int_0^T X_t^2 \, dt, \tag{198}$$

then the drift of the effective process becomes (see Exercises):

$$\hat{F}_{k(a)}(x) = -\frac{\sigma^2}{2a} x. \tag{199}$$

In this case, the paths leading to $A_T = a$ still follow an Ornstein–Uhlenbeck process, but with a modified friction coefficient rather than a re-centering, as illustrated in Fig. 12. This applies to all $a > 0$ away from the typical value $a^* = \sigma^2/(2\gamma)$; for this particular value, $\hat{F}_{k(a^*)}(x) = \hat{F}_{k=0}(x) = F(x) = -\gamma x$, so the original is unchanged, as expected.

The effective process can also be defined for Markov chains with a discrete number of states evolving in continuous or discrete time. In continuous time, the original transition rates $W_{ij}$ of the jump process $X_t$ are modified to

$$(W_k)_{ij} = \frac{1}{r_k(i)} W_{ij} e^{kg(i,j)} r_k(j). \tag{200}$$

This applies to the general observable defined in (125) in terms of the functions $f(x)$ and $g(x,y)$, which depend again on the application considered, and follows by inserting the tilted generator of the process, shown in (129), in the expression of the generalised Doob transform in (186). On the other hand, for a Markov chain in discrete time, the effective process is a modified Markov chain with the transition probabilities

$$(\Pi_k)_{ij} = \frac{1}{(r_k)_i} \frac{(\hat{\Pi})_{ij}}{e^{\lambda(k)}} (r_k)_j = \frac{1}{(r_k)_i} \frac{\Pi_{ij} e^{kf(i)}}{e^{\lambda(k)}} (r_k)_j. \tag{201}$$

Here, $r_k$ is the right eigenvector of $\hat{\Pi}$ associated with the dominant eigenvalue $\zeta_{\max}(\Pi_k) = e^{\lambda(k)}$, so that $(r_k)_i$ denotes the $i$th component of that vector, and $f(i)$ is the function used in the sample mean, appearing in (111). Note that this result does not follow from the generalised Doob transform presented in (186), but from a variant of that transform defined at the level of transition probabilities rather than generators (see Appendix E of [40]).

**Example 13.** Consider the symmetric Bernoulli Markov chain, studied in Example 9, with the sample mean $S_n$ as the observable, so that $f(i) = i$ with $g(i,j) = 0$ or, equivalently, $f(i) = 0$ with $g(i,j) = j$. In this case, it can be checked as an exercise that the effective Markov chain associated with the conditioning of $S_n$ is a non-symmetric Bernoulli Markov chain with transition matrix

$$\Pi_k = \begin{pmatrix} 1-\alpha_k & \alpha_k \\ \beta_k & 1-\beta_k \end{pmatrix}, \tag{202}$$

where $\alpha_k$ and $\beta_k$ are the new jump probabilities (to be determined). The interpretation of this result is that fluctuations of $S_n$ at the level of the symmetric Markov chain arise, effectively, as if the transition probabilities were modified as above. Consequently, the state and transition statistics of sequences $X_1, \ldots, X_n$ selected on the basis of $S_n = s$ should be close to those of the modified Markov chain and not those of the original Markov chain, unless $s$ is the typical value $s^* = p$ associated with $k = 0$.

In the end, it is interesting to note that the notion of an effective process can also be used for describing the large deviations of an IID sequence $X_1, \ldots, X_n$ of RVs. For this application, we already wrote in (184) the joint distribution of $X_1, \ldots, X_n$ conditioned on a value $S_n = s$ of the sample mean and noted there that this distribution is not itself IID. However, following the results just described for Markov processes, we have that the conditioned sequence is asymptotically equivalent to a modified IID sequence $\hat{X}_1, \ldots, \hat{X}_n$, whose distribution is

$$P_k(x) = \frac{e^{kf(x)} P(x)}{E[e^{kf(X)}]}. \tag{203}$$

This means that observations on the sequence of RVs $X_1, \ldots, X_n$ that are conditioned on $S_n = s$ can be described approximately as if they came or were created from a different sequence of IID RVs, playing the role of the effective process, having a different distribution $P_k$. This distribution is called the *exponential tilting* of $P(x)$ and obviously defines a probability distribution, since it is positive and normalised.

This result for IID RVs can be derived in many ways. One left as an exercise follows by noting that an IID sequence is a particular case of discrete-time Markov chain in which the

transition probability $\Pi_{ij}$ does not depend on the starting state $i$, so that $\Pi_{ij} = P(i)$. In this case, the effective Markov chain with the transition probabilities in (201) must also describe an IID sequence of RVs distributed according to $P_k$, as above.

**Example 14.** For our last example, we go back to our first example on the Gaussian sample mean for which the rate function is given by (15), while the SCGF is given by (47). With these two results, the tilted distribution in (203) is found with $f(x) = x$ to be another Gaussian having a different mean:

$$p_k(x) = C\,e^{-\frac{(x-\mu_k)^2}{2\sigma^2}}\,, \qquad \mu_k = \lambda'(k) = \mu + \sigma^2 k\,, \tag{204}$$

$C$ being a normalisation constant. Fixing the value of $k$ to account for the constraint $S_n = s$ with the SCGF then gives

$$p_{k(s)}(x) = C\,e^{-\frac{(x-s)^2}{2\sigma^2}} = p(x-s)\,. \tag{205}$$

This is similar to what we found for the Ornstein–Uhlenbeck process conditioned on the linear integral of $X_t$ (see Example 11) for the simple reason that this process is a Gaussian process. The interpretation of $P_{k(s)}$ is also similar to what we discussed for the Ornstein–Uhlenbeck example. Imagine that we generate many sequences $X_1, \ldots, X_n$ according to $p = \mathcal{N}(\mu, \sigma^2)$ and select the sequences such that $S_n = s$. Then we should see that, as $n$ gets larger, the selected sequences approximately have the same statistics as an IID sequence $\hat{X}_1, \ldots, \hat{X}_n$ distributed according to $p_{k(s)} = \mathcal{N}(s, \sigma^2)$. In particular, the mean of the new sequence $\hat{X}_1, \ldots, \hat{X}_n$ becomes asymptotically close to $s$, matching the mean of the sequences selected in the conditioning.

## 6.3 Asymptotic equivalence

The previous examples show how the effective process is obtained in practice and how it is interpreted as the process that realises the conditioned process $X_t|A_T = a$ as a Markov process. We emphasise again that the conditioned process is not Markovian for a fixed observation or terminal time $T$, because of the global constraint $A_T = a$, but becomes Markovian in an *asymptotic* and *approximate way* as $T \to \infty$. In this last section, we explain what we mean by "asymptotically and approximately close" by listing a number of results known about the effective and conditioned processes, showing that they are asymptotically equivalent as stochastic processes. As in the last section, we state these results without derivation to focus on their physical or practical meaning. References for the results are given in each point below, as well as in the Further reading section.

- **Path distribution equivalence:** The effective and conditioned processes are asymptotically equivalent to one another in the long-time limit at the level of their path distributions, in the sense that

$$\hat{P}[x] \asymp P^a[x]\,. \tag{206}$$

We mentioned this result before. The asymptotic equivalence notation means that

$$\lim_{T \to \infty} \frac{1}{T} \ln \frac{\hat{P}[x]}{P^a[x]} = 0\,, \tag{207}$$

so the effective and conditioned processes have the same path distribution up to sub-exponential corrections in time. This is explained in more detail in [40].

- **Relative entropy equivalence:** Taking the expectation of the asymptotic limit above with respect to the effective process gives

$$\lim_{T \to \infty} \frac{1}{T} D(\hat{P}||P^a) = 0\,, \tag{208}$$

where

$$D(\hat{P}||P^a) = E_{\hat{P}}\left[\ln\frac{\hat{P}[X]}{P^a[X]}\right] = \int \mathcal{D}[x]\,\hat{P}[x]\ln\frac{\hat{P}[x]}{P^a[x]}, \tag{209}$$

is the relative entropy or Kullback–Leibler distance between $\hat{P}$ and $P^a$, expressed as a path integral. In this sense, the effective process can be seen as the Markov process closest to $P^a$, according to the relative entropy distance, that achieves the constraint $A_T = a$ as a concentration point in the limit $T \to \infty$ (see [41]).

- **Observable equivalence:** The observable $A_T$ in the conditioned process is strictly equal to the conditioning value $a$, and so does not fluctuate by definition of this process. In the effective process, we do not have $A_T = a$ in a strict sense, but $A_T \to a$ in probability as $T \to \infty$. Thus, although this process has paths leading to $A_T \neq a$, it mostly realises probabilistically paths such that $A_T = a$, as in the conditioned process. This is similar to the canonical ensemble becoming equivalent to the microcanonical ensemble in the thermodynamic limit: the canonical ensemble has energy fluctuations, but those fluctuations become exponentially unlikely, so most of the microstates of this ensemble have a "fixed" energy, as in the microcanonical ensemble [39].

- **Density and current equivalence:** The effective and conditioned processes have the same stationary distribution, $p_k^*(x)$, and stationary current, $J_k^*(x)$, in the limit $T \to \infty$ (see Exercises). This can be used to define the effective process in a different way as a Markov process with modified density and current consistent with or leading to $A_T = a$. For more details on this characterisation, related to the so-called level 2.5 of large deviations, we refer to [41] and the references listed in the Further reading section.

## 6.4 Exercises

- Show that $p_k^* = r_k l_k$ is the stationary distribution of the effective process. That is, from the definition of $L_k$ show that $L_k^\dagger p_k = 0$. What is the corresponding stationary current $J_k^*(x)$?

- Use the expression of $p_k^*$ to show that the typical value of $A_T$ for this process satisfies $a = \lambda'(k)$. For simplicity, consider observables with $g = 0$. Source: Sec. 5.4 of [40].

- Fill in the calculations of Examples 11 and 12 by finding the effective process associated with the Ornstein–Uhlenbeck process for the two observables considered in those examples. Source: Sec. 6 of [40].

- Derive the results presented in Example 13 on the effective Markov chain associated with the sample mean of the Bernoulli Markov chain.

- Derive the distribution $P_k$ of the effective IID sequence associated with an IID sequence $X_1, \ldots, X_n$ conditioned on a sample mean value $S_n = s$. From the result, complete Example 14 by deriving the expression of $P_k$ and $P_{k(s)}$ from the large deviations of the Gaussian sample mean found in Sec. 2.2.

- Extra: Let $X_t$ be a Markov jump process or a diffusion in continuous time. Show that the conditioning of $X_t$ over $[0, T]$ on the final value $X_T = x_f$ defines a Markov process and derive its generator. Hint: See Sec. 4.2 of [40].

- Extra: Consider the process $(X_t, A_t)$ obtained by adjoining the evolution of the observable in time to the state of the process. Considering the case where $X_t$ is a diffusion, show that this joint process is a Markov process and derive the Fokker–Planck equation

for the joint probability density $p(x, a, t) = P(X_t = x, A_t = a)$. Then show that the conditioning $(X_t, A_t)|A_T = a$ defines a Markov process. How is this conditioning different from the one considered in this section? In other words, in what sense can we say that $X_t|A_T = a$ is not Markovian if $(X_t, A_t)|A_T = a$ is Markovian?

## 6.5 Further reading

- Markov processes conditioned on large deviations and theory of the effective process: [39–41]. Related works: [42–44] and introductory section of [40] for more.

- Applications of the effective process (very limited and partial selection): Density-type observables [45]; current-type observables [46]; linear SDEs [35].

- Variational representation of the effective process: [41] and [47].

- Density and current representation of the effective process using the so-called level 2.5 of large deviations: [41].

- Level 2.5 of large deviations: [48].

- Large deviation estimation and simulations: [4], [50], [49] and references therein.

- Brownian and Markov bridges: [51] and forthcoming [52].

## Acknowledgments

We thank Abhishek Dhar, Joachim Krug, Satya Majumdar, Alberto Rosso, and Grégory Schehr for their work and dedication in organising a most pleasant and interesting school and for the opportunity to take part in it.

**Funding information** Our presence at the school was made possible with funding from the following institutions and organisations: Sorbonne Université (INB), Stellenbosch University, NITheCS, and the NRF (DWHC), LMU München (VK), Stellenbosch University and Université de Grenoble (HT).

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
