# Peer review of "An introduction to large deviations with applications in physics"

_SciPost Physics Lecture Notes, doi:SciPost Phys. Lect. Notes 104 (2025)_

## Round 1 · Referee Report · Anonymous (Referee 1) · 2025-5-1

Strengths

1-pedagogical effort to explain both the mathematical and the physical aspects 2-many examples 3-many exercices 4-many pointers for further reading 5- the links with the other lectures of the same school are explained in section II-A

Weaknesses

1- As stated in the introduction, the link with the video recordings [1] is not straightforward : « The notes are not a transcript of the lectures, as such, but go through all the material that was presented in each of the six lectures… We arranged the notes by topics rather than by lectures » So a natural question of students or newcomers in the field would be : « Should I look at the videos first or should I read the notes directly ? »

2-The relations with the previous lectures notes [3,4,5] by the same lecturer are not explained clearly in the Introduction while natural questions of students or newcomers in the field would be : « What lectures should I read ? In what order ? Are some more introductory or more advanced ? Are some more general or more specialized ? Which one is better for my goals ?»

Report

These lectures notes will be useful to students and newcomers
in the field of large deviations, so I recommend the publication.
My main concern is the expression
« the large deviation approximation or large deviation principle » that appear many times,
and that can be very puzzling, since the word ‘approximation’ and the word ‘principle’ are clearly not at all synonymous !
The word ‘approximation’ is explained in Eqs 19-20 containing o(n), although I feel that most physicists would instead emphasize that Eq. 21 is the ‘exact asymptotic behavior’, and would prefer to keep the word ‘approximation’ for the many other situations where one is not able to compute exactly the leading order for large N.
As far as I can tell, the authors define the expression ‘large deviation principle’ in Eqs 22 and 25, but they do not explain at all in what sense it is supposed to be a ‘principle’, while they make great efforts to explain many other words.
Indeed in physics, the word ‘principle’ has the meaning of a ‘physical law that has to be postulated in addition to the mathematical framework in order to make the link with the physical world’ , since it was used by Newton in his « Philosophiæ Naturalis Principia Mathematica »,
by Einstein with the « principle of relativity », by Dirac in his « Principles of Quantum Mechanics » , etc… So for physicists, it can be confusing to use to word ‘principle’ to comment Eq 22, as if it were a new physical theory with respect to the standard statistical physics.
But of course I also understand that ‘LDP’ is a standard expression in mathematics,
so the authors are free to decide whether they wish to add explanations
or not on the use of the word ‘principle’.

Requested changes

1) Typo in the sentence following the sentence containing Eq. 57 : I think that the authors wanted to write « provided that the SCGF of X exists and is differentiable .. »

2) In contrast to the clarity of the whole manuscript, the two sentences at the bottom of p22 « Moreover, for applying large deviation theory to Markov chains, we do not need to assume, strictly speaking, that Π is ergodic. This is a convenience more than a necessary requirement. » are very obscur and ambiguous, especially the expressions « strictly speaking » and « convenience more than a necessary .. ». So I think that the authors should better explain what they really mean.

3) Typo : I think that there is a missing square in Eq 169 that should read \int dx \psi^2(x)=1 in order to recover the standard normalization of the square of the wave-function in quantum mechanics.

Recommendation

Publish (surpasses expectations and criteria for this Journal; among top 10%)

  • validity: top
  • significance: top
  • originality: good
  • clarity: top
  • formatting: perfect
  • grammar: perfect

Author:  Daniël Cloete  on 2025-08-29  [id 5762]

(in reply to Report 1 on 2025-05-01)

We thank the referee for their careful reading of the manuscript and for the constructive suggestions.
We have made the three requested changes, and clarified our use of the term 'principle' in the context of the large deviation principle.

---

## Round 1 · Referee Report · Anonymous (Referee 2) · 2025-5-5

Strengths

The authors illustrate key concepts with numerous examples and include ample references for further reading.

Weaknesses

Naturally, there is an overlap with the 2009 Physics Report article by the lecturer.

Report

These lecture notes provide a thorough overview of the topic of large deviations. While there is some overlap with the lecturer's earlier review article, the notes also discuss recent material that was not covered in the previous work.

Recommendation

Publish (surpasses expectations and criteria for this Journal; among top 10%)

  • validity: top
  • significance: top
  • originality: good
  • clarity: high
  • formatting: excellent
  • grammar: excellent

Author:  Daniël Cloete  on 2025-08-29  [id 5761]

(in reply to Report 2 on 2025-05-05)

We thank the referee for their positive report and for taking the time to review the lecture notes.
We are glad that they found the work clear and suitable for publication. No changes were requested.

---

## Round 3 · Author Response

We thank the referees for their thoughtful and constructive reports.
For Referee 1, we have made the three requested changes and clarified our use of the term principle in the context of the large deviation principle.
For Referee 2, no changes were required, as they found the manuscript clear and suitable for publication.
We believe that the revised version addresses all comments and is now suitable for publication.

---

## Round 3 · List of Changes

Corrected minor typos following Eq. (57) and in Eq. (169).
Clarified the discussion on the ergodicity of Π.
Added a footnote to Definition 1 explaining our use of the term principle.

---

## Editorial Decision

published